# M-Attack-V2: Pushing the Frontier of Black-Box LVLM Attacks via Fine-Grained Detail Targeting

## Abstract

Black-box adversarial attacks on Large Vision–Language Models (LVLMs) present unique challenges due to the absence of gradient access and complex multimodal decision boundaries. While prior `M-Attack` demonstrated notable success in exceeding 90% attack success rate on GPT-4o/o1/4.5 by leveraging local crop-level matching between source and target data, we show this strategy introduces high-variance gradient estimates. Specifically, we empirically find that gradients computed over randomly sampled local crops are nearly orthogonal, violating the implicit assumption of coherent local alignment and leading to unstable optimization. To address this, we propose a theoretically grounded **_gradient denoising_** framework that redefines the adversarial objective as an expectation over local transformations. Our first component, *Multi-Crop Alignment (MCA)*, estimates the expected gradient by averaging gradients across diverse, independently sampled local transformations. This manner significantly reduces gradient variance, thus enhancing convergence stability. Recognizing an asymmetry in the roles of source and target transformations, we also introduce *Auxiliary Target Alignment (ATA)*. ATA regularizes the optimization by aligning the adversarial example not only with the primary target image but also with auxiliary samples drawn from a semantically correlated distribution. This constructs a smooth semantic trajectory in the embedding space, acting as a low-variance regularizer over the target distribution. Finally, we reinterpret prior momentum as replay through the lens of local matching as variance-minimizing estimators under the crop-transformed objective landscape. Momentum replay stabilizes and amplifies transferable perturbations by maintaining gradient directionality across local perturbation manifolds. Together, MCA, ATA, momentum replay, and a delicately selected ensemble set constitute `M-Attack-V2`, a principled framework for robust black-box LVLM attack. Empirical results show that our framework improves the attack success rate on Claude-4.0 (✳️) from **8%→30%**, on Gemini-2.5-Pro (🔷) from **83%→97%**, and on on GPT-5 (🌀) from **98%→100%**, significantly surpassing all existing black-box LVLM attacking methods.

## 1 Introduction

Large Vision-Language Models (LVLMs) have become foundational to modern AI systems, enabling multimodal tasks like image captioning (Hu et al., 2022; Salaberria et al., 2023; Chen et al., 2022b; Tschannen et al., 2023), VQA (Luu et al., 2024; Özdemir & Akagündüz, 2024), and visual reasoning (OpenAI, 2025). However, their visual modules remain vulnerable to adversarial attacks, subtle perturbations that mislead models while remaining imperceptible to humans. Prior efforts, including AttackVLM (Zhao et al., 2023), CWA (Chen et al., 2024), SSA-CWA (Dong et al., 2023a), AdvDiffVLM (Guo et al., 2024), and most effectively, `M-Attack` (Li et al., 2025), which have exploited this weakness through local-level matching and surrogate model ensembles, surpassing 90% success rates on models like GPT-4o.

Despite its effectiveness, our analysis reveals that `M-Attack`'s gradient signals are highly unstable: Even overlapping large pixel regions, two consecutive local crops share *nearly orthogonal gradients*.

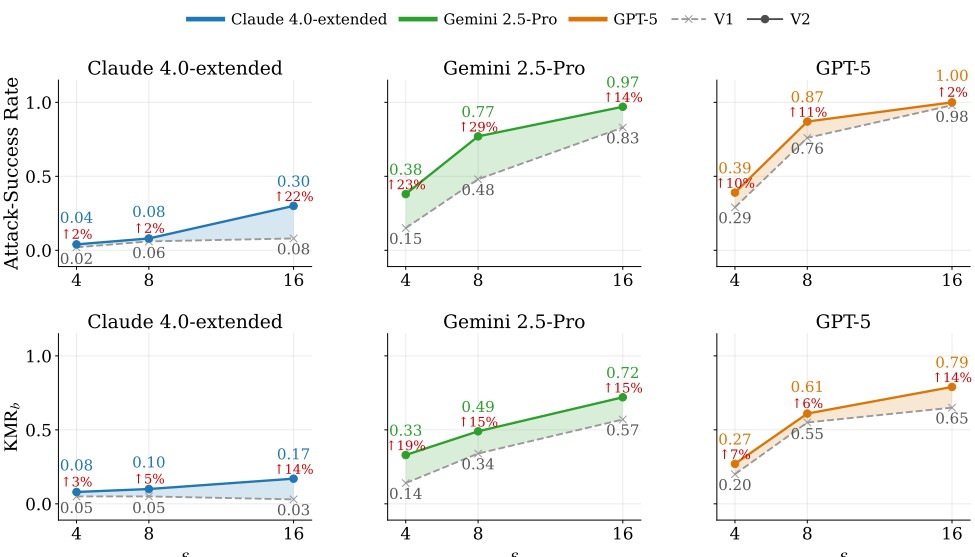

Figure 1: Improvement of `M-Attack-V2` over `M-Attack` on up-to-date commercial black-box models(Claude-4.0-extended, Gemini 2.5-Pro and GPT-5)

In other words, high similarity in pixel and embedding space does not translate to high similarity in gradient space. The reason is that ViTs' gradient pattern is sensitive to translation. A tiny shift changes pixels contained in each token, altering self-attention. Moreover, patch-wise, spike-like gradient amplifies the mismatch within just a few pixels. We counter this effect by aggregating gradients from multiple crops within the same iteration, a strategy we call *Multi-Crop Alignment* (MCA). From a theoretical angle, MCA aggregates gradients across multiple views in a single iteration, smoothing local inconsistencies and improving cross-crop gradient stability.

We further observe that the source and target transformations in `M-Attack` operate in different semantic spaces: one emphasizing extraction, the other generalization. Aggressive target augmentation introduces harmful variance. Our *Auxiliary Target Alignment* (ATA) mitigates this by identifying semantically similar auxiliary images to create a low-variance embedding subspace, then applying only mild shifts to enhance transferability without destabilizing the optimization.

Classic momentum is reinterpreted under this framework as *Patch Momentum* (PM), a replay mechanism that recycles past gradients across random crops to stabilize optimization. In parallel, we also re-examine and enrich `M-Attack`'s model selection criterion and choose a delicately selected ensemble set with diverse patch sizes to mitigate the difficulty in cross-patch transfer, of which we find that the attention concentrates more on the main object. We term it *Patch Ensemble*[+] (PE[+]).

Together, these components, MCA, ATA, PM, and PE[+], form the basis of `M-Attack-V2`, a robust gradient denoising framework that significantly outperforms existing black-box attack methods. Our method raises attack success rates from 98%→100% on GPT-5, 8%→30% on Claude-4, and 83%→97% on Gemini-2.5-Pro, achieving state-of-the-art performance across the board. This study not only offers a practical, modular attack strategy but also sheds light on the gradient behavior of ViT-based LVLMs under local perturbations. We hope these insights will drive further research into transferable adversarial optimization under realistic black-box constraints.

## 2 BACKGROUND

**Large Vision Language Models.** Transformer-based LVLMs learn visual-semantic representations from large-scale image-text data, enabling tasks like image captioning (Salaberria et al., 2023; Hu et al., 2022; Chen et al., 2022b; Tschannen et al., 2023), visual QA (Luu et al., 2024; Özdemir & Akagündüz, 2024), and cross-modal reasoning (Wu et al., 2025; Ma et al., 2023; Wang et al., 2024). Open-source models such as BLIP-2 (Li et al., 2022), Flamingo (Alayrac et al., 2022), and LLaVA (Liu et al., 2023) show strong benchmark performance. Commercial models like GPT-4o, Claude-3.5 (Anthropic, 2024a), and Gemini-2.0 (Team et al., 2023) offer advanced reasoning and

real-world adaptability, with their successors, GPT-o3 (OpenAI, 2025), Claude 3.7-Sonnet (Anthropic, 2024b), and Gemini-2.5-Pro, able to reason in the text modality and vision modality.

**LVLM transfer-based attack.** Black-box attacks include query-based (Dong et al., 2021; Ilyas et al., 2018) and transfer-based (Dong et al., 2018; Liu et al., 2017) methods; this work focuses on the latter. AttackVLM (Zhao et al., 2023) introduced transfer-based targeted attacks on LVLMs using CLIP (Radford et al., 2021) and BLIP (Li et al., 2022) as surrogates, showing that image-to-image feature matching outperforms cross-modal optimization, a strategy adopted by later works (Chen et al., 2024; Guo et al., 2024; Dong et al., 2023a; Li et al., 2025). CWA (Chen et al., 2024) and SSA-CWA (Dong et al., 2023a) applied this principle to commercial models like Bard (Team et al., 2023), with CWA enhancing transferability via sharpness-aware minimization (Foret et al., 2021; Chen et al., 2022a), and SSA-CWA introducing spectrum-guided augmentation via SSA (Long et al., 2022). AnyAttack (Zhang et al., 2024) utilizes image-image matching through large-scale pertaining and a subsequent fine-tuning. AdvDiffVLM (Guo et al., 2024) embeds feature matching into diffusion guidance, introduces Adaptive Ensemble Gradient Estimation (AEGE) for smoother ensemble scores. Notably, M-Attack significantly outperforms these methods through a simple yet effective local-level matching framework with an ensemble of diverse patch sizes. Building upon this framework, FOA-Attack (Jia et al., 2025) introduces Feature Optimal Alignment, extending alignment from the CLS token to local patch tokens in embedding space, yielding further improvements. However, the local-level matching framework itself has notable limitations. Before analyzing and addressing them, we briefly introduce the necessary background of the local-level matching

**Local-level matching in M-Attack.** Consider a clean source image $\tilde{\mathbf{X}}_{\text{sou}}$ and a target image $\mathbf{X}_{\text{tar}}$. The objective of black-box transfer attacks is to minimally perturb the source image by $\delta$ so that the perturbed image $\mathbf{X}_{\text{sou}} = \tilde{\mathbf{X}}_{\text{sou}} + \delta$ aligns semantically with the target under an inaccessible black-box model $f_\xi$. Due to the inaccessibility of $f_\xi$, surrogate models $f_\phi$ approximate the semantic alignment via cosine similarity (CS):

$$\arg\max_{\mathbf{X}_{\text{sou}}} \text{CS}(f_\phi(\mathbf{X}_{\text{sou}}), f_\phi(\mathbf{X}_{\text{tar}})) \quad \text{s.t.} \quad \|\delta\|_p \leq \epsilon. \tag{1}$$

M-Attack enhances Eq. (1) using *local-level matching*. At iteration $i$, it applies predefined local transformations $\mathcal{T}_s$ and $\mathcal{T}_t$ to extract local area $\hat{\mathbf{x}}_i^s$ from the source $\mathbf{X}_{\text{sou}}$ and $\hat{\mathbf{x}}_i^t$ from the target $\mathbf{X}_{\text{tar}}$, respectively. These transformations satisfy essential properties, such as spatial overlap and diversified coverage of extracted local regions $\{\hat{\mathbf{x}}_i\}$ (Li et al., 2025). Formally, the local-level matching optimizes:

$$\mathcal{M}_{\mathcal{T}_s, \mathcal{T}_t} = \mathbb{E}_{f_{\phi_j} \sim \phi}[\text{CS}(f_{\phi_j}(\hat{\mathbf{x}}_i^s), f_{\phi_j}(\hat{\mathbf{x}}_i^t))], \tag{2}$$

where $f_{\phi_j}$ is sampled from an ensemble of surrogate models $\phi$. Intuitively, matching local image regions instead of entire images enhances the semantic precision of perturbations by directing optimization towards semantically significant details. Despite its effectiveness, M-Attack encounters a critical challenge of *unexpectedly low* gradient similarity, which we investigate in detail next.

## 3 METHOD

### 3.1 LIMITATIONS OF LOCAL-LEVEL MATCHING IN M-Attack

**Extremely low gradient overlap.** In M-Attack two random crops $\hat{\mathbf{x}}_i^s$ and $\hat{\mathbf{x}}_i^t$ are matched at every iteration. One would expect the gradients inside the shared region of two successive source crops $(\hat{\mathbf{x}}_i^s, \hat{\mathbf{x}}_{i+1}^s)$ to correlate, because the underlying pixels partly coincide. Supursingly, Fig. 2b shows the opposite: their cosine similarity is ***almost zero***. We then keep one crop fixed and vary the other across scales and IoUs (Fig. 2a). Our finding reveals an exponential decay that plateaus below 0.1 once the overlap is smaller than 0.80 IoU.

**Source.** We find two main reasons behind this high variance: ViT's inherent sensitivity to translation and overlooked asymmetry within the local matching framework. We discuss them below.

*Patch-wise, spike-like gradient sensitive to translation.* Because ViTs tokenize images on a fixed, non-overlapping grid, even sub-pixel changes each patch's token mix. These token changes ripple through self-attention, altering weights and redirecting gradients for *all* tokens, so the resulting pixel-level gradient pattern diverges sharply. Worse, gradient magnitudes are uneven. Therefore, even similar patterns but missing a few pixels might break gradient similarity (Fig. 3b).

*Asymmetric Transform Branches.* In M-Attack, both the *source* and *target* images are cropped, yet

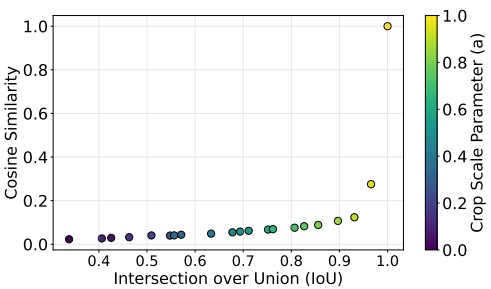 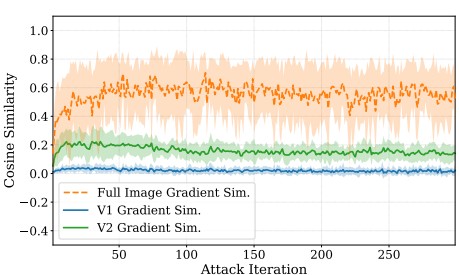

(a) Similarity over IoU. The results are averaged from 20 runs with different crop parameter $a$ for $[a, 1.0]$.

(b) Comparison of gradient similarity from full image update and local matching over each iteration

Figure 2: Similarities of gradients from different crops. a) similarity over IoU for different crops by fixing in one iteration; b) similarity between two consecutive gradients across iterations. Results are averaged from 200 runs.

playing distinct roles. Cropping the source acts directly in *pixel space*: it rearranges patch embeddings and attention weights in the forward pass, ending up with guidance of different views. By contrast, cropping the target sorely translate the target representation, thereby shifting the reference embedding in *feature space*. One sculpts the perturbation, while another moves the goalpost, formulating asymmetric matching. M-Attack overlooked this and implementations target translation alternate between a *radical* crop and an identity map, struggles between explore-exploitation trade-off and potentially risk in high variance of target embedding.

**Asymmetric Matching over Expectation.** To mitigate the issues above, we begin by concisely reformulating the original objective function as an expectation over local transformations within an asymmetric matching framework:

$$\min_{\|\mathbf{X}_{\text{sou}}\|_p \leq \epsilon} \mathbb{E}_{\mathcal{T} \sim \mathcal{D}, y \sim \mathcal{Y}} \left[ \mathcal{L}(f(\mathcal{T}(\mathbf{X}_{\text{sou}})), \mathbf{y}) \right], \tag{3}$$

where $\mathcal{D}$ represents the distribution of local transformations, and $\mathcal{Y}$ denotes the distribution over target semantics. $\|\cdot\|_p$ is $\ell_p$ constraint for imperceptibility. Conceptually, this formulation corresponds to embedding specific semantic content $y$ into a locally transformed area $\mathcal{T}(\mathbf{X}_{\text{sou}})$, thus highlighting the intrinsic asymmetry compared to M-Attack's original formulation. Within this framework, our proposed enhancements, i.e., *Multi-Crop Alignment* (MCA) and *Auxiliary Target Alignment* (ATA), can be interpreted as strategies to improve the accuracy of the expectation estimation and the sampling quality of the semantic distribution $\mathcal{Y}$.

### 3.2 *Gradient Denoising* VIA MULTI-CROP ALIGNMENT (MCA)

To obtain a low-variance estimate of the expected loss gradient $\mathbb{E}_{\mathcal{T} \sim \mathcal{D}, y \sim \mathcal{Y}} \left[ \nabla_{\mathbf{X}_{\text{sou}}} \mathcal{L}(f(\mathcal{T}(\mathbf{X}_{\text{sou}})), \mathbf{y}) \right]$, we draw $K$ independent crops $\{\mathcal{T}\}_{k=1}^{K}$ and average their individual gradients:

$$\nabla_{\mathbf{X}_{\text{sou}}} \hat{\mathcal{L}}(\mathbf{X}_{\text{sou}}) = \frac{1}{K} \sum_{k=1}^{K} \nabla_{\mathbf{X}_{\text{sou}}} \mathcal{L}(f(\mathcal{T}_k(\mathbf{X}_{\text{sou}})), \mathbf{y}). \tag{4}$$

This *Multi-Crop Alignment* is an unbiased Monte-Carlo estimator, reducing the variance with $K > 1$.

**Theorem 1.** *Let* $g_k = \nabla_{\mathbf{X}_{sou}} \mathcal{L}(f(\mathcal{T}_k(\mathbf{X}_{sou})), y)$ *denote the gradient from* $\mathcal{T}_k$, $\mu = \mathbb{E}[g_k], \sigma^2 = \mathbb{E}[\|g_k - \mu\|_2^2]$ *denote the mean and variance, and* $p_{k\ell}$ *denote the pair-wise correlation* $p_{k\ell} = \frac{\langle g_k - \mu, g_\ell - \mu \rangle}{\|g_k - \mu\|^2 \|g_\ell - \mu\|^2}$. *The gradient variance from* $K$ *averaged crops is bounded by*

$$\text{Var}\left( \frac{1}{K} \sum_{k=1}^{K} g_k \right) \leq \frac{\sigma^2}{K} + \frac{K-1}{K} \overline{p} \sigma^2, \tag{5}$$

*where* $\overline{p} = \mathbb{E}[p_{kl}], \ k \neq \ell$ *is the expectation of pair-wise correlation*

All crops share the same underlying image, so $\overline{p} \neq 0$. The ideal $\sigma^2/K$ decay is therefore tempered by the correlation term $\overline{p}\sigma^2$. Empirically, averaging a modest number ($K = 10$) of almost-orthogonal

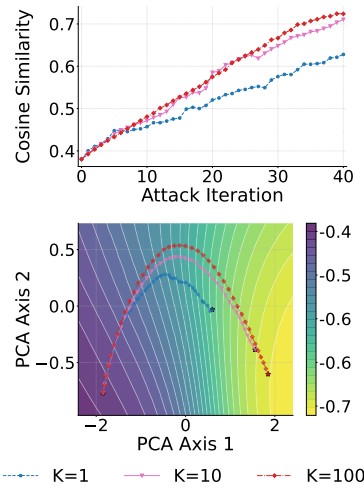

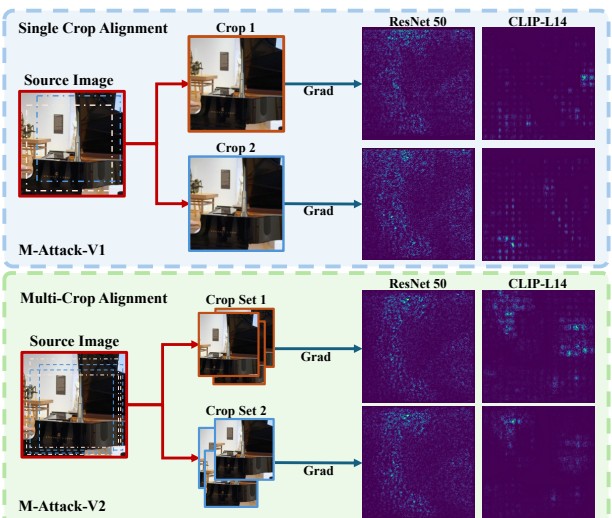

(a) Comparison of optimization trajectories with different $K$, $K = 1$ refers to single crop alignment.

(b) Gradient pattern between different crop strategies in `M-Attack` and `M-Attack-V2`.

Figure 3: Comparison of: a) different trajectories against different $K$; b) gradient pattern of single crop alignment against multi-crop alignment (MCA). The gradient pattern of ResNet 50 remains consistent when large pixels are overlapped, while the gradient pattern of ViTs changes dramatically. MCA helps to smooth out this impact.

gradients still yields benefit, since the uncorrelated component of the variance shrinks as $1/K$. Simultaneously, the optimizer leverages multiple diverse transformations per update, with minimal interference among almost orthogonal gradients. Fig. 3a illustrates an accelerated convergence with $K = 10$, with margin improvement provided by $K = 100$.

This averaging also alleviates the known translation sensitivity of ViTs. As shown in Fig. 3b, using two crop sets yields noticeably higher gradient consistency than the single-crop alignment in `M-Attack`. In MCA, high-activity regions remain stable (upper left and center right), while the single-crop case shifts focus from center right to lower left. As a result, gradient similarity across iterations increases from near zero in `M-Attack` to around 0.2 (Fig. 2b).

### 3.3  *Improved Sampling Quality* VIA AUXILIARY TARGET ALIGNMENT (ATA)

Selecting a representative target embedding $y \in \mathcal{Y}$ is challenging because the underlying distribution $\mathcal{Y}$ is not observable. `M-Attack` mitigates this by seeding at the unaltered target embedding $f(\mathbf{X}_{\text{tar}})$ and exploring its vicinity with transformed views $f(\mathcal{T}_t(\mathbf{X}_{\text{tar}}))$ thereby sketching a locally semantic manifold that serves as a proxy for $\mathcal{Y}$. However, the exploration–exploitation trade-off remains problematic. *Radical* transformations leap too far, dragging $y$ outside the genuine target region; *conservative* transformations, while semantically faithful, barely shift the embedding, leaving the optimization starved of informative signal.

To stabilize this process, we introduce $P$ auxiliary images $\{\mathbf{X}_{\text{aux}}^{(p)}\}_{p=1}^{P}$ that act as additional anchors, collectively forming a richer sub-manifold of aligned embeddings. During each update, we apply a *mild* random transformation $\tilde{\mathcal{T}} \sim \tilde{\mathcal{D}}$ to every anchor, nudging the ensemble in a coherent yet restrained manner and thus providing low-variance, information-rich gradients for optimization. Let $y_0 = f(\hat{\mathcal{T}}_0(\mathbf{X}_{\text{tar}}))$, $\tilde{y}_p = f(\tilde{\mathcal{T}}_p(\mathbf{X}_{\text{aux}}^{(p)}))$ denote sampled semantics in one iteration. The objective $\hat{\mathcal{L}}$ in Equ. (4) becomes

$$\hat{\mathcal{L}} = \frac{1}{K} \sum_{k=1}^{n} \left[ \mathcal{L}(f(\mathcal{T}_k(\mathbf{X}_{\text{sou}})), y_0) + \frac{\lambda}{P} \sum_{p=1}^{P} \mathcal{L}(f(\mathcal{T}_k(\mathbf{X}_{\text{sou}})), \tilde{y}_p) \right] \tag{6}$$

where $\lambda \in [0, 1]$ interpolates between the original target and its auxiliary neighbors. $\lambda = 0$ reduce to `M-Attack` local-local matching with single target. ATA trade-off exploration (auxiliary diversity) and exploitation (main-target fidelity), providing low-variance, semantics-preserving updates. The

auxiliary set can be built variously, i.e., through image-image retrieval or diffusion methods. We now theoretically analyze ATA with three mild assumptions:

*Assumption 3.1 (*Lipschitz surrogate*). Surrogate $f$ is $L$-continuous: $\|f(y) - f(x)\| \le L\|y - x\|$.*

*Assumption 3.2 (*Bounded Auxiliary Data*). For auxiliary data $\mathbf{X}_{aux}^{(p)}$ retrieved via semantic similarity to a target $\mathbf{X}_{tar}$, we have: $\mathbb{E}[\|f(\mathbf{X}_{aux}^{(p)}) - f(\mathbf{X}_{tar})\|] \le \delta$ (justification in Appdix. C.3).*

*Assumption 3.3 (*Bounded transformation*). Random transformation $\mathcal{T} \sim D_\alpha$ has bounded pixel-level distortion: $\mathbb{E}[\|\mathcal{T}(\mathbf{X}) - \mathbf{X}\|] \le \alpha$*

**Theorem 2.** *Let $\mathcal{T} \sim D_\alpha$ denote the transformation used in* M-Attack, *and $\tilde{\mathcal{T}} \sim D_{\tilde{\alpha}}$ with $\tilde{\alpha} \ll \alpha$ the transformation in* M-Attack-V2. *Define **embedding drift** of transformation $\mathcal{T}$ applied to $\mathbf{X}$ on model $f$ as: $\Delta_{\mathrm{drift}}(\mathcal{T}; \mathbf{X}) := \mathbb{E}_{\mathcal{T}}[\|f(\mathcal{T}(\mathbf{X})) - f(\mathbf{X}_{tar})\|]$. Then, we have:*

$$\Delta_{\mathrm{drift}}(\mathcal{T}; \mathbf{X}_{\mathrm{tar}}) \le L\alpha, \qquad \Delta_{\mathrm{drift}}(\tilde{\mathcal{T}}; \mathbf{X}_{\mathrm{aux}}^{(p)}) \le L\tilde{\alpha} + \delta. \tag{7}$$

Specifically, the term $L\alpha$ captures the inherent asymmetry caused by transformations in pixel space, necessitating the multiplier $L$ to map pixel-level perturbations into embedding-space effects. In contrast, the auxiliary data *directly operates* in embedding space, leading to a manageable bound $\delta$. Practically, estimating $\delta$ is notably easier than estimating $L\alpha$. Lower $\delta$ inherently indicates better semantic alignment, allowing M-Attack-V2 to operate effectively under reduced distortion ($\tilde{\alpha} \ll \alpha$). Thus, ATA strategically allocates its shift budget towards more meaningful exploration through $\delta$, achieving a sweet point between exploration and exploitation.

**Cost.** Each iteration back-propagates through the $K$ source crops and only forward-propagates the $P$ auxiliary targets. Since a backward pass is roughly twice as expensive as a forward pass, the per-iteration complexity is $\mathcal{O}(K(3 + P))$, doubling overhead when $P = 3$.

### 3.4 PATCH MOMENTUM WITH BUILT-IN REPLAY EFFECT

Momentum, introduced in MI-FGSM (Dong et al., 2018), is widely adopted for transferability. Define the momentum buffer as: $m_r = \beta_1 m_{r-1} + (1 - \beta_1)\nabla_{\hat{\mathbf{x}}^s}\hat{\mathcal{L}}_r(\hat{\mathbf{x}}^s)$, where $\beta_1 \in [0, 1)$ is the first-order momentum coefficient and $\nabla_{\hat{\mathbf{x}}^s}\hat{\mathcal{L}}_r(\hat{\mathbf{x}}^s)$ is our MCA-ATA-estimated gradient $g_r$ at iteration $r$.

Under the local-matching view, this mechanism can be reinterpreted as formulating a streaming MCA to enforce temporal consistency across gradient directions in the space of random crops. Unrolling the EMA for pixel $k$ exposes an alternative interpretation:

$$m_i(k) = (1 - \beta) \sum_{j=0}^{i} \beta^j \mathbf{1}\{k \in M_{i-j}\} g_{i-j}(k), \tag{8}$$

where $M_i$ denotes the pixel indices included in iteration $i$, $m_i(k)$ and $g_i(k)$ respectively denotes momentum and gradient for pixel $k$. Each crop involving pixel $k$ is therefore replayed in future iterations with geometrically decaying weight, allowing rarely sampled regions (such as corners) to persist long enough to combat the gradient starvation. Spike-shaped gradients are further moderated by the Adam-style (Kingma & Ba, 2017) second moment, $v_r = \beta_2 v_{r-1} + (1-\beta_2)g_r^2$, whose scaling effect is essential in our empirical study. The momentum does not directly improve gradient similarity but continuously re-injects historical crops across patches, effectively maintaining gradient directionality across local perturbation manifolds. We therefore term it *Patch Momentum* to distinguish.

The whole procedure, combining MCA, ATA, and PM, is detailed in Alg. 1. We use a different color to differentiate between M-Attack-V2 and M-Attack. We use PGD (Madry et al., 2018) with ADAM (Kingma & Ba, 2017) for line 12. Appx. F.2 presents analogous results for variants.

## 4 EXPERIMENTS

### 4.1 EXPERIMENTAL SETUP

**Metrics.** We adopt the evaluation protocol of M-Attack, reporting the *Attack Success Rate* (ASR) via *GPTScore* and the *Keywords Matching Rate* (KMR) at three thresholds $\{0.25, 0.5, 1.0\}$, denoted

---

**Algorithm 1** `M-Attack-V2`

---

**Require:** clean image $\mathbf{X}_{\text{clean}}$; primary target $\mathbf{X}_{\text{tar}}$; auxiliary set $\mathcal{A} = \{\mathbf{X}_{\text{aux}}^{(p)}\}_{p=1}^{P}$; patch ensemble$^{+}$
$\quad \Phi^{+} = \{\phi_j\}_{j=1}^{m}$; iterations $n$, step size $\alpha$, perturbation budget $\epsilon$; number of crops $K$, auxiliary weight
$\quad \lambda \ (0 \le \lambda \le 1)$;
1: $\mathbf{X}_{\text{adv}} \leftarrow \mathbf{X}_{\text{clean}}$,
2: **for** $i = 1$ **to** $n$ **do**
3: $\quad$ Draw $K$ transforms $\{\mathcal{T}_k\}_{k=1}^{K} \sim \mathcal{D}, g \leftarrow \mathbf{0}$
4: $\quad$ **for** $k = 1$ **to** $K$ **do** $\qquad\qquad\qquad\qquad\qquad$ ▷ — crop loop (vectorizable) —
5: $\quad\quad$ Draw $\{\tilde{\mathcal{T}}_p\}_{p=0}^{P} \sim \tilde{D}$
6: $\quad\quad$ **for** $j = 1$ **to** $m$ **do**
7: $\quad\quad\quad$ $y_0 = f(\tilde{\mathcal{T}}_p(\mathbf{X}_{\text{tar}})), \ y_p = f(\tilde{\mathcal{T}}_p(\mathbf{X}_{\text{aux}}^{(p)})), p = 1, \ldots, P$ $\quad$ ▷ Transform target and auxiliary data
8: $\quad\quad\quad$ Compute $\hat{\mathcal{L}}_k = (f_{\phi_j}(\mathcal{T}_k(\mathbf{X}_{\text{sou}})), y_0) + \frac{\lambda}{P}\sum_{p=1}^{P} \mathcal{L}(f_{\phi_j}(\mathcal{T}_k(x)), \tilde{y}_p)$
9: $\quad\quad\quad$ $g \leftarrow g + \frac{1}{Km}\nabla_{\mathbf{X}_{\text{sou}}}\hat{\mathcal{L}}_k$
10: $\quad\quad$ **end for**
11: $\quad$ **end for**
12: $\quad$ Updated $\mathbf{X}_{\text{adv}}$ based on $g$ with Patch Momentum
13: **end for**
14: **return** $\mathbf{X}_{\text{adv}}$

---

as $\text{KMR}_a$, $\text{KMR}_b$, and $\text{KMR}_c$ (Li et al., 2025). KMR measures semantic alignment using human-annotated keywords, considering a match successful if the rate exceeds threshold $x$. The evaluation prompt and keyword sets follow `M-Attack` exactly.

**Surrogate candidates.** We follow surrogate selections from prior ensemble-based methods (Zhang et al., 2024; Dong et al., 2023a; Guo et al., 2024; Li et al., 2025). Our candidate pool covers CLIP variants (CLIP-B/16, B/32, L/14, CLIP$^{\dagger}$-G/14, CLIP$^{\dagger}$-B/32, CLIP$^{\dagger}$-H/14, CLIP$^{\dagger}$-B/16, CLIP$^{\dagger}$-BG/14), DinoV2 (Oquab et al., 2023) (Small, Base, Large), and BLIP-2 (Li et al., 2023).

**Victim models and dataset.** We evaluate state-of-the-art commercial MLLMs: GPT-4o/o3/5, Claude-3.7/4.0 (extended), and Gemini-2.5-Pro-Preview (Team et al., 2023). Clean images are drawn from the *NIPS 2017 Adversarial Attacks and Defenses Competition* dataset (K et al., 2017). Following SSA-CWA (Dong et al., 2023b) and `M-Attack` (Li et al., 2025), we randomly sample 100 images, retrieving auxiliary sets from the COCO training set (Lin et al., 2015) using CLIP-B/16 embedding similarity. Further results on a 1k image subset are in the Appx. F.1. Additional Results on open-source LLMs are in the Appx. F.3. We provide the Huggingface identifiers of the model in Appx B. All the BLIP2 (Li et al., 2023) variants on Huggingface share the same vision encoder. Therefore, we use only one. The *milder target transformation* includes random resized crop ($[0.9, 1.0]$), random horizontal flip ($p = 0.5$), and random rotation ($\pm 15°$).

**Hyperparameters.** Unless noted, perturbations are bounded by $\ell_{\infty}$ with $\epsilon = 16$ and optimized for 300 steps. We set the step size to $\alpha = 0.75$ for Claude and $\alpha = 1.0$ for all other methods, mirroring `M-Attack`. For `M-Attack-V2`, $\alpha = 1.275$, $\beta_1 = 0.9, \beta_2 = 0.99$ for momentum, $K = 10$, $P = 2$, and $\lambda = 0.3$ for MCA and ATA. Ablation on $\alpha$ is in Appx E.1, with $\beta, K, P, \lambda$ in Appx E.2.

### 4.2 Extensive Evaluation Across LVLMs and Settings

**Transferability across LVLMs.** Tab. 1 illustrates the superiority of our `M-Attack-V2` compared to the other black-box LVLM attack method. Our method leads others by a large margin, including `M-Attack`. On GPT-5 our `M-Attack-V2` even achieves 100% ASR and 97% ASR on Gemini-2.5, with ASR on Claude 4.0-extended further improved by 22%, which is almost impossible for `M-Attack`

| Method | Model | GPT-5 | | | | Claude 4.0-thinking | | | | Gemini 2.5-Pro | | | | Imperceptibility | |
|---|---|---|---|---|---|---|---|---|---|---|---|---|---|---|---|
| | | $\text{KMR}_a$ | $\text{KMR}_b$ | $\text{KMR}_c$ | ASR | $\text{KMR}_a$ | $\text{KMR}_b$ | $\text{KMR}_c$ | ASR | $\text{KMR}_a$ | $\text{KMR}_b$ | $\text{KMR}_c$ | ASR | $\ell_1\downarrow$ | $\ell_2\downarrow$ |
| | **B/16** | 0.08 | 0.03 | 0.02 | 0.05 | 0.03 | 0.00 | 0.00 | 0.00 | 0.08 | 0.04 | 0.00 | 0.00 | 0.034 | 0.040 |
| AttackVLM (Zhao et al., 2023) | **B/32** | 0.07 | 0.05 | 0.04 | 0.02 | 0.03 | 0.03 | 0.00 | 0.01 | 0.09 | 0.05 | 0.00 | 0.02 | 0.036 | 0.041 |
| | **Laion$^{\dagger}$** | 0.02 | 0.01 | 0.00 | 0.03 | 0.02 | 0.01 | 0.00 | 0.00 | 0.09 | 0.05 | 0.00 | 0.01 | 0.035 | 0.040 |
| AdvDiffVLM (Guo et al., 2024) | **Ensemble** | 0.04 | 0.02 | 0.01 | 0.01 | 0.04 | 0.01 | 0.01 | 0.01 | 0.03 | 0.01 | 0.00 | 0.00 | 0.064 | 0.095 |
| SSA-CWA (Dong et al., 2023a) | **Ensemble** | 0.08 | 0.04 | 0.00 | 0.08 | 0.03 | 0.02 | 0.01 | 0.05 | 0.05 | 0.03 | 0.01 | 0.08 | 0.059 | 0.060 |
| AnyAttack (Zhang et al., 2024) | **Ensemble** | 0.09 | 0.03 | 0.00 | 0.06 | 0.05 | 0.03 | 0.00 | 0.01 | 0.35 | 0.06 | 0.01 | 0.34 | 0.048 | 0.052 |
| FOA-Attack (Jia et al., 2025) | **Ensemble** | 0.90 | 0.67 | 0.23 | 0.94 | 0.13 | 0.09 | 0.00 | 0.13 | 0.61 | 0.80 | 0.15 | 0.86 | 0.031 | 0.036 |
| M-Attack (Li et al., 2025) | **Ensemble** | 0.89 | 0.65 | 0.25 | 0.98 | 0.12 | 0.03 | 0.00 | 0.08 | 0.81 | 0.57 | 0.15 | 0.83 | **0.030** | **0.036** |
| `M-Attack-V2` (Ours) | **Ensemble** | **0.92** | **0.79** | **0.30** | **1.00** | **0.27** | **0.17** | **0.04** | **0.30** | **0.87** | **0.72** | **0.22** | **0.97** | 0.038 | 0.044 |

Table 1: Comparison on three target LVLMs. $^{\dagger}$: pre-trained on LAION (Schuhmann et al., 2022).

| $\epsilon$ | Method | GPT-4o | | | | Claude 3.7-thinking | | | | Gemini 2.5-Pro | | | | Imperceptibility | |
|---|---|---|---|---|---|---|---|---|---|---|---|---|---|---|---|
| | | $KMR_a$ | $KMR_b$ | $KMR_c$ | ASR | $KMR_a$ | $KMR_b$ | $KMR_c$ | ASR | $KMR_a$ | $KMR_b$ | $KMR_c$ | ASR | $\ell_1\downarrow$ | $\ell_2\downarrow$ |
| 4 | AttackVLM (Zhao et al., 2023) | 0.08 | 0.04 | 0.00 | 0.02 | 0.04 | 0.01 | 0.00 | 0.00 | 0.10 | 0.04 | 0.00 | 0.01 | 0.010 | 0.011 |
| | SSA-CWA (Dong et al., 2023a) | 0.05 | 0.03 | 0.00 | 0.03 | 0.04 | 0.01 | 0.00 | 0.02 | 0.04 | 0.01 | 0.00 | 0.04 | 0.015 | 0.015 |
| | AnyAttack (Zhang et al., 2024) | 0.07 | 0.02 | 0.00 | 0.05 | 0.05 | **0.05** | **0.02** | **0.06** | 0.05 | 0.02 | 0.00 | 0.10 | 0.014 | 0.015 |
| | M-Attack (Li et al., 2025) | 0.30 | 0.16 | 0.03 | 0.26 | 0.06 | 0.01 | 0.00 | 0.01 | 0.24 | 0.14 | 0.02 | 0.15 | **0.009** | **0.010** |
| | M-Attack-V2 (Ours) | **0.59** | **0.34** | **0.10** | **0.58** | **0.06** | 0.02 | 0.00 | 0.02 | **0.48** | **0.33** | **0.07** | **0.38** | 0.012 | 0.013 |
| 8 | AttackVLM (Zhao et al., 2023) | 0.08 | 0.02 | 0.00 | 0.01 | 0.04 | 0.02 | 0.00 | 0.01 | 0.07 | 0.01 | 0.00 | 0.01 | 0.020 | 0.022 |
| | SSA-CWA (Dong et al., 2023a) | 0.06 | 0.02 | 0.00 | 0.04 | 0.04 | 0.02 | 0.00 | 0.02 | 0.02 | 0.00 | 0.00 | 0.05 | 0.030 | 0.030 |
| | AnyAttack (Zhang et al., 2024) | 0.17 | 0.06 | 0.00 | 0.13 | 0.07 | 0.07 | 0.02 | 0.05 | 0.12 | 0.04 | 0.00 | 0.13 | 0.028 | 0.029 |
| | M-Attack (Li et al., 2025) | 0.74 | 0.50 | 0.12 | 0.82 | 0.12 | 0.06 | 0.00 | 0.09 | 0.62 | 0.34 | 0.08 | 0.48 | **0.017** | **0.020** |
| | M-Attack-V2 (Ours) | **0.87** | **0.69** | **0.20** | **0.93** | **0.23** | **0.14** | **0.02** | **0.22** | **0.72** | **0.49** | **0.21** | **0.77** | 0.023 | 0.023 |
| 16 | AttackVLM (Zhao et al., 2023) | 0.08 | 0.02 | 0.00 | 0.02 | 0.01 | 0.00 | 0.00 | 0.01 | 0.03 | 0.01 | 0.00 | 0.00 | 0.036 | 0.041 |
| | SSA-CWA (Dong et al., 2023a) | 0.11 | 0.06 | 0.00 | 0.09 | 0.06 | 0.04 | 0.01 | 0.12 | 0.05 | 0.03 | 0.01 | 0.08 | 0.059 | 0.060 |
| | AnyAttack (Zhang et al., 2024) | 0.44 | 0.20 | 0.04 | 0.42 | 0.19 | 0.08 | 0.01 | 0.22 | 0.35 | 0.06 | 0.01 | 0.34 | 0.048 | 0.052 |
| | M-Attack (Li et al., 2025) | 0.82 | 0.54 | 0.13 | 0.95 | 0.31 | 0.21 | 0.04 | 0.37 | 0.81 | 0.57 | 0.15 | 0.83 | **0.030** | **0.036** |
| | M-Attack-V2 (Ours) | **0.91** | **0.78** | **0.40** | **0.99** | **0.56** | **0.32** | **0.11** | **0.67** | **0.87** | **0.72** | **0.22** | **0.97** | 0.038 | 0.044 |

Table 2: Ablation study on the impact of perturbation budget ($\epsilon$).

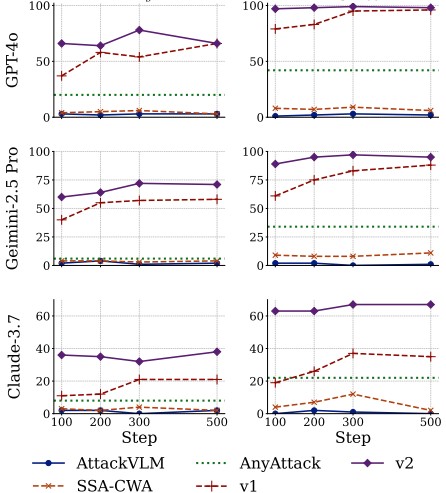

| Component | | | Gemini 2.5-Pro | | | | Claude 3.7-extended | | | |
|---|---|---|---|---|---|---|---|---|---|---|
| MCA | ATA | PM | $KMR_a$ | $KMR_b$ | $KMR_c$ | ASR | $KMR_a$ | $KMR_b$ | $KMR_c$ | ASR |
| | | | 0.87 | 0.72 | 0.22 | 0.97 | 0.56 | 0.32 | 0.11 | 0.67 |
| ✗ | | | 0.85 ↓0.02 | 0.70 ↓0.02 | 0.21 ↓0.01 | 0.92 ↓0.05 | 0.52 ↓0.04 | 0.35 ↑0.03 | 0.08 ↓0.03 | 0.66 ↓0.01 |
| | ✗ | | 0.85 ↓0.02 | 0.68 ↓0.04 | 0.21 ↓0.01 | 0.93 ↓0.04 | 0.55 ↓0.01 | 0.22 ↓0.10 | 0.10 ↓0.01 | 0.62 ↓0.05 |
| ✗ | ✗ | | 0.82 ↓0.05 | 0.62 ↓0.10 | 0.22 – | 0.93 ↓0.04 | 0.44 ↓0.12 | 0.31 ↓0.01 | 0.08 ↓0.03 | 0.62 ↓0.05 |
| | | ✗ | 0.82 ↓0.05 | 0.71 ↓0.01 | 0.21 ↓0.01 | 0.96 ↓0.01 | 0.52 ↓0.04 | 0.32 ↓0.00 | 0.10 ↓0.01 | 0.66 ↓0.01 |

Table 3: Effect of removing each component. Numbers below each value denote the change relative to the full model (first row). ✗ marks the component(s) disabled.

| Model | $KMR_a$ | $KMR_b$ | $KMR_c$ | ASR |
|---|---|---|---|---|
| GPT-o3 (o3-2025-04-16) | 0.91 | 0.71 | 0.23 | 0.98 |

Table 4: Results of M-Attack-V2 on vision reasoning model

Figure 4: Comparison of different methods under different step budgets.

to attack. There is also a notable improvement on the KMR, indicating that our method generates a perturbation that targets the semantics more effectively, thus more recognizable by the target black-box model. Note that these improvements are accompanied by a slight increase in the perturbation norms for $l_1$ and $l_2$. Previous $l_1$ and $l_2$ norms are caused by insufficient optimization through near-orthogonal gradients. Our M-Attack-V2 mitigates this issue, exploring more sufficiently inside the $l_\infty$ ball. Thus, it slightly increases the magnitude of the perturbation with a neglectable impact on the actual visual effect. See the Appx. G.1 for visualizations of adversarial samples.

**Performance under budgets.** Tab. 2 compares performance across varying perturbation budgets ($\epsilon$). Our method consistently ranks among the top two across all settings, achieving notably large margins when outperforming competitors, highlighting its effectiveness in exploring within different $\ell_\infty$ balls. Fig. 4 further compares performance under varying optimization budgets (total steps). Our method converges faster, approaching optimal results within 300 steps, whereas M-Attack requires an additional 200 steps, suggesting slower convergence. At fewer steps (100 and 200), M-Attack exhibits a notable performance drop, while our method maintains stable ASR and $KMR_b$. This robustness arises from reduced variance compared to M-Attack, which is more sensitive to random cropping and aggressive target transformations, necessitating additional iterations to stabilize.

**Robustness Against Vision-Reasoning Models.** Reasoning in text modality does not extend to alter information from the vision backbone. Instead, we further evaluate M-Attack-V2 against GPT-o3, a model enhanced with visual reasoning capabilities. As shown in Tab. 4, GPT-o3 exhibits slightly better robustness than GPT-4o. However, the limited improvement suggests that its reasoning module is not explicitly trained to detect adversarial manipulations in the image. Thus, even after reasoning, GPT-o3 remains susceptible to M-Attack-V2. Reasoning process is presented in Appx G.2.

Figure 5: Comparison of two types of attention maps. Left: attention map that sparsely separates in different regions; right: attention map that focus to the main object.

| Surrogate | C−L/14 | C†−L/14 | D−S/14 | D−B/14 | D−L/14 | C−B/16 | C†−B/16 | C−B/32 | C†−B/32 | BLIP2 | Avg/14 | Avg/16 | Avg/32 | Avg/All |
|---|---|---|---|---|---|---|---|---|---|---|---|---|---|---|
| C−L/14 | N/A | 0.40 | 0.10 | 0.13 | 0.12 | 0.45 | 0.40 | 0.34 | 0.24 | 0.48 | 0.25 | 0.42 | 0.29 | 0.30 |
| C†−L/14 | 0.44 | N/A | 0.24 | 0.24 | 0.21 | 0.55 | 0.57 | 0.37 | 0.33 | 0.61 | 0.35 | 0.56 | 0.35 | 0.39 |
| D−S/14 | 0.25 | 0.39 | N/A | 0.45 | 0.38 | 0.41 | 0.45 | 0.32 | 0.25 | 0.46 | 0.39 | 0.43 | 0.28 | 0.37 |
| D−B/14 | 0.29 | 0.36 | 0.33 | N/A | 0.51 | 0.37 | 0.39 | 0.31 | 0.23 | 0.47 | 0.39 | 0.38 | 0.27 | 0.36 |
| D−L/14 | 0.26 | 0.31 | 0.12 | 0.32 | N/A | 0.34 | 0.30 | 0.21 | | 0.42 | 0.29 | 0.33 | 0.26 | 0.29 |
| C−B/16 | 0.44 | 0.43 | 0.21 | 0.18 | 0.13 | N/A | 0.53 | 0.37 | 0.27 | 0.51 | 0.32 | 0.53 | 0.32 | 0.34 |
| C†−B/16 | 0.43 | 0.51 | 0.22 | 0.21 | 0.15 | 0.57 | N/A | 0.39 | 0.34 | 0.52 | 0.34 | 0.57 | 0.36 | 0.37 |
| C−B/32 | 0.37 | 0.43 | 0.21 | 0.11 | 0.09 | 0.55 | 0.53 | N/A | 0.49 | 0.46 | 0.28 | 0.54 | 0.49 | 0.36 |
| C†−B/32 | 0.31 | 0.49 | 0.27 | 0.18 | 0.12 | 0.53 | 0.61 | 0.58 | N/A | 0.50 | 0.31 | 0.57 | 0.58 | 0.40 |
| BLIP2 | 0.39 | 0.43 | 0.15 | 0.20 | 0.26 | 0.45 | 0.43 | 0.33 | 0.25 | N/A | 0.29 | 0.44 | 0.29 | 0.32 |

Table 5: Comparison of embedding transferability over 1k images. MCA/ATA excluded to show standalone performance. C/D = CLIP/DinoV2. Gray denotes selected models.

### 4.3 ABLATION STUDY

**Selection of surrogate model.** Ensembling surrogate models is typical for enhancing black-box adversarial transferability. To further improve, advanced gradient aggregation methods (Zhang et al., 2024; Guo et al., 2024) have been proposed; yet another practical and efficient approach, parallel to aggregation, is to select models strategically. We first profile the embedding transferability on different surrogate models, presented in Tab. 5. Results show that cross-model, especially cross-patchsize transfer, is difficult. Therefore, we retain models with diverse patch sizes that perform well in Tab 5. Trials in the appendix yield our *Patch Ensemble+* (PE+), comprising *CLIP†-G/14, CLIP-B/16, CLIP-B/32, and CLIP†-B/32*. Attention maps reveal a possible explanation: PE+ models tend to concentrate attention on the main object, whereas others exhibit dispersed focus across unrelated regions. We hypothesize that focusing on the main object enhances transferability, as all models share the common objective of identifying core semantic content. In contrast, attention to scattered regions may capture model-specific biases that do not generalize well across architectures.

**Ablation on remaining components.** Tab. 3 isolates the effect of each module beyond PE+ (GPT-4o omitted due to neglectable differences). On both Gemini-2.5-Pro and Claude-3.7-extended, activating MCA or ATA alone delivers ∼5% gains on average, most visible in ASR and $KMR_b$, with consistent improvements on $KMR_a$/$KMR_c$. Removing PM yields only a minor drop, suggesting it is complementary rather than fundamental. Overall, MCA and ATA constitute the principal variance-reduction mechanisms, while PM functions as a low-cost memory that extends the effective momentum horizon with a biased gradient, further suppressing variance and adding robustness.

## 5 CONCLUSION

We find that `M-Attack` suffers from unstable gradients and identify the root causes as high variance and overlooked asymmetric matching. To this end, we introduce a principled framework that includes Multi-Crop Alignment (MCA) for variance reduction, Auxiliary Target Alignment (ATA) for semantic consistency, and Patch Momentum (PM) for replay-based stabilization. Combined with a refined surrogate model ensemble (PE+), these components form `M-Attack-V2`, which achieves state-of-the-art results across multiple black-box LVLMs. We hope this study provides practical insights and encourages further research into stable and transferable adversarial optimization under realistic black-box constraints.

ETHICS STATEMENT

This study investigates adversarial attacks on black-box LVLMs. Such work is inherently dual-use: methods designed to evaluate and improve model robustness might also be misused to circumvent safety mechanisms or generate harmful outputs. Nevertheless, our primary intent is to highlight vulnerabilities and encourage the development of more robust LVLMs. To mitigate potential risks associated with our research, we take several precautions: 1) We exclusively utilize publicly available datasets (e.g., COCO, NIPS 2017 Adv. Attacks) that contain no personally identifiable information and do not involve human participants. 2) We avoid targeting any production systems, do not interact with private user data or protected services, and rigorously adhere to the respective service providers' terms of use. 3) We release all code, prompts, and generated adversarial examples to facilitate research into model robustness and defense mechanisms. All authors acknowledge and adhere to the ICLR Code of Ethics.

REPRODUCIBILITY STATEMENT

We are committed to ensuring complete reproducibility of our results. The paper clearly defines our objectives and algorithms (Algorithm 1), optimization strategies and hyperparameters (Section 4.1, Appendix B), settings for ablation studies (Appendices E.1, E.2, Section 4.3), as well as precise evaluation protocols, metrics, and model specifications (Appendix B). Additionally, an *anonymous* repository containing scripts to reproduce all key tables and figures, along with controlled seeds, environment setup files (e.g., `requirements.txt`), and detailed YAML configuration files for each experiment, is provided during the review process and will be released.

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

# APPENDIX

## CONTENTS

## A   COMPLEMENTARY DETAILS OF `M-Attack-V2`

Alg. 2 and Alg. 3 provide detailed update rule of line 13 in Alg. 1. Fig. 6 provides a comparison between the entire procedure of `M-Attack` and `M-Attack-V2` under the local-matching framework. Notably, `M-Attack` utilizes a radical crop on the target image, risking unrelated or broken semantics for the source image to align. Our ATA anchors more points inside the semantic manifold (blue), and provides a mild transformation to provide a coherence sampling from the target semantic manifold.

---

**Algorithm 2** `M-Attack-V2` (Adam variant)

---

**Require:** clean image $\mathbf{X}_{\text{clean}}$; primary target $\mathbf{X}_{\text{tar}}$; auxiliary set $\mathcal{A} = \{\mathbf{X}_{\text{aux}}^{(p)}\}_{p=1}^{P}$; patch ensemble$^+$ $\Phi^+ = \{\phi_j\}_{j=1}^{m}$; iterations $n$, step size $\alpha$, perturbation budget $\epsilon$; Adam $\beta_1, \beta_2, \eta$; number of crops $K$, auxiliary weight $\lambda$;

1: $\mathbf{X}_{\text{adv}} \leftarrow \mathbf{X}_{\text{clean}}, m \leftarrow 0, v \leftarrow 0$
2: **for** $i = 1$ **to** $n$ **do**
3:     Draw $K$ transforms $\{\mathcal{T}_k\}_{k=1}^{K} \sim \mathcal{D}$
4:     $g \leftarrow 0$                                                 ▷ accumulate over crops
5:     **for** $k = 1$ **to** $K$ **do**                             ▷ — crop loop —
6:         Draw $\{\tilde{\mathcal{T}}_p\}_{p=0}^{P} \sim \tilde{\mathcal{D}}$
7:         **for** $j = 1$ **to** $m$ **do**
8:             $y_0 = f(\tilde{\mathcal{T}}_0(\mathbf{X}_{\text{tar}}))$
9:             $y_p = f(\tilde{\mathcal{T}}_p(\mathbf{X}_{\text{aux}}^{(p)})),\ p = 1, \ldots, P$
10:            $\hat{\mathcal{L}}_k = \mathcal{L}\big(f_{\phi_j}(\mathcal{T}_k(\mathbf{X}_{\text{adv}})), y_0\big) + \frac{\lambda}{P}\sum_{p=1}^{P}\mathcal{L}\big(f_{\phi_j}(\mathcal{T}_k(\mathbf{X}_{\text{adv}})), y_p\big)$
11:            $g \leftarrow g + \frac{1}{Km}\nabla_{\mathbf{X}_{\text{adv}}}\hat{\mathcal{L}}_k$
12:         **end for**
13:     **end for**                                        ▷ — Adam update —
14:     $m \leftarrow \beta_1 m + (1 - \beta_1)g$
15:     $v \leftarrow \beta_2 v + (1 - \beta_2)g^{\odot 2}$
16:     $\hat{m} \leftarrow m/(1 - \beta_1^i);\ \hat{v} \leftarrow v/(1 - \beta_2^i)$
17:     $\mathbf{X}_{\text{adv}} \leftarrow \text{clip}_{\mathbf{X}_{\text{clean}}, \epsilon}\big(\mathbf{X}_{\text{adv}} + \alpha\,\hat{m}/(\sqrt{\hat{v}} + \eta)\big)$
18: **end for**
19: **return** $\mathbf{X}_{\text{adv}}$

---

**Algorithm 3** `M-Attack-V2` (MI-FGSM variant)

---

**Require:** clean image $\mathbf{X}_{\text{clean}}$; primary target $\mathbf{X}_{\text{tar}}$; auxiliary set $\mathcal{A} = \{\mathbf{X}_{\text{aux}}^{(p)}\}_{p=1}^{P}$; patch ensemble$^+$ $\Phi^+ = \{\phi_j\}_{j=1}^{m}$; iterations $n$, step size $\alpha$, perturbation budget $\epsilon$; momentum decay $\gamma$; number of crops $K$, auxiliary weight $\lambda$;

1: $\mathbf{X}_{\text{adv}} \leftarrow \mathbf{X}_{\text{clean}}, \mu \leftarrow 0$
2: **for** $i = 1$ **to** $n$ **do**
3:     Draw $K$ transforms $\{\mathcal{T}_k\}_{k=1}^{K} \sim \mathcal{D}$
4:     $g \leftarrow 0$
5:     **for** $k = 1$ **to** $K$ **do**
6:         Draw $\{\tilde{\mathcal{T}}_p\}_{p=0}^{P} \sim \tilde{\mathcal{D}}$
7:         **for** $j = 1$ **to** $m$ **do**
8:             $y_0 = f(\tilde{\mathcal{T}}_0(\mathbf{X}_{\text{tar}}))$
9:             $y_p = f(\tilde{\mathcal{T}}_p(\mathbf{X}_{\text{aux}}^{(p)})),\ p = 1, \ldots, P$
10:            $\hat{\mathcal{L}}_k = \mathcal{L}\big(f_{\phi_j}(\mathcal{T}_k(\mathbf{X}_{\text{adv}})), y_0\big) + \frac{\lambda}{P}\sum_{p=1}^{P}\mathcal{L}\big(f_{\phi_j}(\mathcal{T}_k(\mathbf{X}_{\text{adv}})), y_p\big)$
11:            $g \leftarrow g + \frac{1}{Km}\nabla_{\mathbf{X}_{\text{adv}}}\hat{\mathcal{L}}_k$
12:         **end for**
13:     **end for**                                    ▷ — MI-FGSM update —
14:     $\mu \leftarrow \gamma\,\mu + \dfrac{g}{\|g\|_1}$
15:     $\mathbf{X}_{\text{adv}} \leftarrow \text{clip}_{\mathbf{X}_{\text{clean}}, \epsilon}\big(\mathbf{X}_{\text{adv}} + \alpha\,\text{sign}(\mu)\big)$
16: **end for**
17: **return** $\mathbf{X}_{\text{adv}}$

---

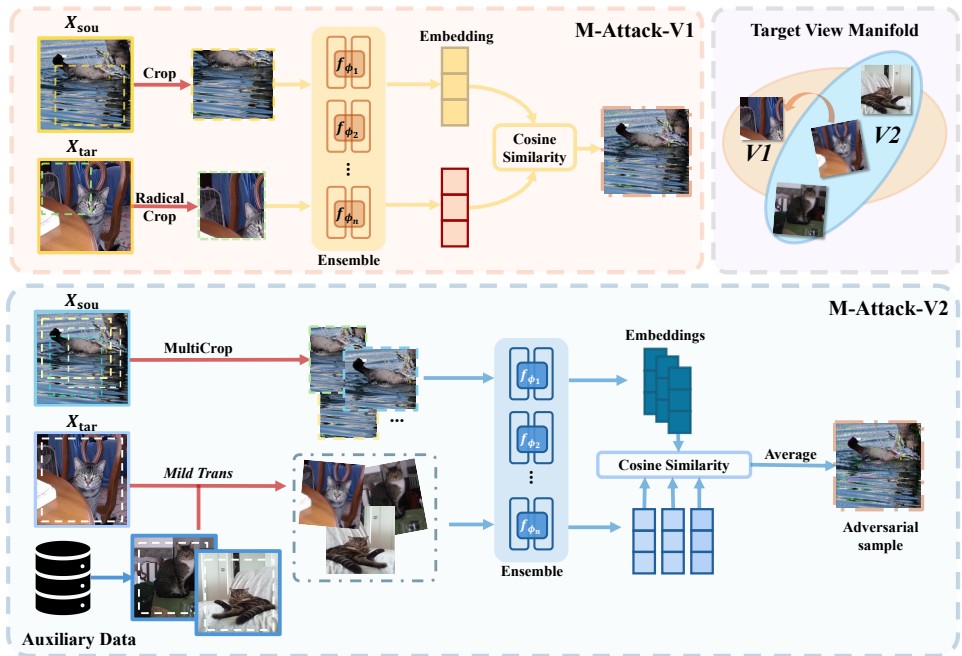

Figure 6: Comparison of one step between `M-Attack` and `M-Attack-V2`.

## B  COMPLEMENTARY DETAILS OF EXPERIMENTAL SETUP

The experiment's seed is 2023. It is conducted on a Linux platform (Ubuntu 22.04) with 6 NVIDIA RTX 4090 GPUs. The temperatures of all LLMs are set to 0. The threshold of the ASR is set to 0.3, following `M-Attack`. Tab. 8 provides a map from model names in this paper to their identifiers in HuggingFace. We use GPT-5-thinking-low (setting reasoning effort to low in the API) for all results in the main paper, with results on other reasoning budgets presented in the Appx. F.4

## C  ADDITIONAL DETAILS FOR THEORETICAL ANALYSIS

### C.1  PROOF FOR THEOREM 1

This section provides detailed proof of the upper bound in Equ. (5). For variance, we have

$$
\mathrm{Var}(\hat{g}_K) := \mathbb{E}\|\hat{g}_K - \mu\|^2
$$

$$
= \mathbb{E}\left\|\frac{1}{K}\sum_{k=1}^{K}(g_k - \mu)\right\|^2
$$

$$
= \frac{1}{K^2}\sum_{k=1}^{K}\sum_{\ell=1}^{K}\mathbb{E}[(g_k - \mu)^\top(g_\ell - \mu)] \tag{9}
$$

$$
= \frac{1}{K^2}\left(\underbrace{\sum_{k=1}^{K}\mathbb{E}\|g_k - \mu\|_2^2}_{K\sigma^2} + 2\underbrace{\sum_{1\le k<\ell\le K}\mathbb{E}[\langle g_k - \mu, g_\ell - \mu\rangle]}_{\text{cross terms}}\right)
$$

The diagonal part is reduced to the mean. We now provide an upper bound for the cross terms. Recall $p_{k\ell} = \frac{\langle g_k - \mu, g_\ell - \mu\rangle}{\|g_k - \mu\|^2\|g_\ell - \mu\|^2}$, we have

$$
\mathbb{E}[\langle g_k - \mu, g_\ell - \mu\rangle] = \mathbb{E}\left[\rho_{k\ell}\|g_k - \mu\|_2\|g_\ell - \mu\|_2\right]. \tag{10}
$$

Since all crops share the same marginal distribution, i.e. $\mathbb{E}\|g_k - \mu\|_2 = \mathbb{E}\|g_\ell - \mu\|_2 = \sigma$, applying the Cauchy-Schwarz inequality to Equ. (10) yields

$$\mathbb{E}[\langle g_k - \mu, g_\ell - \mu \rangle] \leq \mathbb{E}[\rho_{k\ell}]\sqrt{\mathbb{E}\|g_k - \mu\|_2^2}\sqrt{\mathbb{E}\|g_\ell - \mu\|_2^2} = \bar{\rho}\sigma^2, \tag{11}$$

where $\bar{p}$ is $\mathbb{E}[p_{k\ell}], k \neq \ell$. Plugging this into the double sum term yields

$$\sum_{1 \leq k < \ell \leq K} \mathbb{E}\left[\langle g_k - \mu, g_\ell - \mu \rangle\right] \leq \frac{K(K-1)}{2}\bar{\rho}\sigma^2. \tag{12}$$

The $\frac{K(K-1)}{2}$ appears since there are total $\frac{K(K-1)}{2}$ terms for $\sum_{k<\ell}$. Thus substituting Equ. (12) back to the cross item part in the Equ. (9) yields

$$\mathrm{Var}(\hat{g}_K) \leq \frac{1}{K^2}\left(K\sigma^2 + K(K-1)\bar{p}\sigma^2\right) = \frac{1}{K}\sigma^2 + \frac{K-1}{K}\bar{p}\sigma^2 \tag{13}$$

Therefore, we have the upper bound provided in the Sec. 3.2.

## C.2 PROOF OF THEOREM 2

We begin with the drift analysis for `M-Attack`:

$$\begin{aligned}
\Delta_{\mathrm{drift}}(\mathcal{T}; \mathbf{X}_{\mathrm{tar}}) &= \mathbb{E}_{\mathcal{T} \sim D\alpha}[\|f(\mathcal{T}(\mathbf{X}_{\mathrm{tar}})) - f(\mathbf{X}_{\mathrm{tar}})\|] \\
&\leq L \cdot \mathbb{E}_{\mathcal{T} \sim D\alpha}[\|\mathcal{T}(\mathbf{X}_{\mathrm{tar}}) - \mathbf{X}_{\mathrm{tar}}\|] \qquad \text{(Assumption 3.1)} \\
&\leq L\alpha \qquad\qquad\qquad\qquad\qquad\qquad \text{(Assumption 3.3)}.
\end{aligned} \tag{14}$$

Next, we analyze the drift for `M-Attack-V2` using the triangle inequality and the above assumptions:

$$\begin{aligned}
\Delta_{\mathrm{drift}}(\tilde{\mathcal{T}}; \mathbf{X}_{\mathrm{aux}}^{(p)}) &= \mathbb{E}_{\tilde{\mathcal{T}} \sim D_{\tilde{\alpha}}}\left[\|f(\tilde{\mathcal{T}}(\mathbf{X}_{\mathrm{aux}}^{(p)})) - f(\mathbf{X}_{\mathrm{tar}})\|\right] \\
&\leq \mathbb{E}_{\tilde{\mathcal{T}}}\left[\|f(\tilde{\mathcal{T}}(\mathbf{X}_{\mathrm{aux}}^{(p)})) - f(\mathbf{X}_{\mathrm{aux}}^{(p)})\| + \|f(\mathbf{X}_{\mathrm{aux}}^{(p)}) - f(\mathbf{X}_{\mathrm{tar}})\|\right] \qquad \text{(Triangle inequality)} \\
&= \mathbb{E}_{\tilde{\mathcal{T}}}\left[\|f(\tilde{\mathcal{T}}(\mathbf{X}_{\mathrm{aux}}^{(p)})) - f(\mathbf{X}_{\mathrm{aux}}^{(p)})\|\right] + \mathbb{E}\left[\|f(\mathbf{X}_{\mathrm{aux}}^{(p)}) - f(\mathbf{X}_{\mathrm{tar}})\|\right] \\
&\leq L\,\mathbb{E}_{\tilde{\mathcal{T}}}\left[\|\tilde{\mathcal{T}}(\mathbf{X}_{\mathrm{aux}}^{(p)}) - \mathbf{X}_{\mathrm{aux}}^{(p)}\|\right] + \delta \qquad\qquad\qquad \text{(Assumps. 3.1, 3.2)} \\
&\leq L\tilde{\alpha} + \delta \qquad\qquad\qquad\qquad\qquad\qquad\qquad\qquad \text{(Assump. 3.3)}.
\end{aligned}$$

Thus, we have completed the proof of Theorem 2.

## C.3 JUSTIFICATION FOR ASSUMPTION 3.2

Assumption 3.2 is derived from the retrieval mechanism for auxiliary data. Specifically, $X_{\mathrm{aux}}^{(p)}$ represents the $p$-th closest embedding to the target $X_{\mathrm{tar}}$ from a database $\mathcal{D}$, defined explicitly by:

$$\mathbf{X}_{\mathrm{aux}}^{(p)} \in \arg\mathrm{top}_P\left\{\mathbf{X} \in \mathcal{D} : \frac{f(\mathbf{X})^\top f(\mathbf{X}_{\mathrm{tar}})}{|f(\mathbf{X})||f(\mathbf{X}_{\mathrm{tar}})|}\right\}, \tag{15}$$

where $\mathrm{top}_P$ denotes selecting the top-$P$ nearest neighbors according to cosine similarity. Given that embeddings $f(\mathbf{X})$ are typically normalized, semantic closeness naturally bounds the expected distance between $f(\mathbf{X}_{\mathrm{aux}}^{(p)})$ and $f(\mathbf{X}_{\mathrm{tar}})$ by $\delta$, thus validating Assumption 3.2. In such a case, to estimate $\delta$, use $2\left(1 - f(\mathbf{X}_{\mathrm{aux}}^{(P)})^\top f(\mathbf{X}_{\mathrm{tar}})\right)$

## D FULL PROCESS OF SURROGATE MODEL SELECTION

This section details the process of selecting our final ensemble, PE$^+$. Exhaustively testing all model combinations is computationally infeasible, so we employ a heuristic-driven approach. We begin by excluding DiNO-large and BLIP2 due to their poor transferability, as shown in Tab. 5. Our initial experiments focus on evaluating the effectiveness of homogeneous ensembles—comprising models with the same patch size—versus mixed patch size ensembles. Specifically, we construct five ensembles: (1) patch-14 CLIP (CLIP-L/14, CLIP$^\dagger$-G/14), (2) patch-14 DiNOv2 (Dino-base,

Dino-large), (3) patch-16 CLIP (CLIP-B/16, CLIP$^\dagger$-B/16), and (4) patch-32 CLIP (CLIP-B/32, CLIP$^\dagger$-B/32). Results are presented in Tab. 6. These results reveal that the patch-32 CLIP ensemble performs best on Claude 3.7, while GPT-4o and Gemini 2.5 Pro favor models with patch sizes 14 and 16. This supports the findings in Sec. 4.3: although using a fixed patch size can mitigate architectural bias, it still inherits the intrinsic bias of the patch size itself.

To address this, we adopt a cross-patch size strategy. Starting from the patch-32 CLIP ensemble, due to its strong performance on Claude and consistent transferability across patch-16 and patch-32 models. We incrementally incorporate one model each from patch sizes 14 and 16. We evaluate various combinations, with results summarized in Tab. 7. The resulting ensemble, PE$^+$, achieves the most balanced performance, ranking first on 7 metrics and a close second on 3 others, across 12 evaluation metrics.

| Variant | Surrogate Set (2 models) | GPT-4o | | | | Claude 3.7-extended | | | | Gemini 2.5-Pro | | | |
|---|---|---|---|---|---|---|---|---|---|---|---|---|---|
| | | KMR$_a$ | KMR$_b$ | KMR$_c$ | ASR | KMR$_a$ | KMR$_b$ | KMR$_c$ | ASR | KMR$_a$ | KMR$_b$ | KMR$_c$ | ASR |
| Pair$_1$ | Dino-B, Dino-S | 0.84 | 0.57 | 0.15 | 0.91 | 0.09 | 0.04 | 0.00 | 0.05 | **0.84** | 0.53 | 0.11 | 0.81 |
| Pair$_2$ | L16, B/16 | **0.86** | **0.69** | 0.21 | **0.96** | 0.16 | 0.10 | 0.01 | 0.16 | **0.84** | 0.59 | 0.15 | 0.91 |
| Pair$_3$ | L32, B/32 | 0.76 | 0.52 | 0.13 | 0.79 | **0.46** | **0.29** | **0.06** | **0.70** | 0.58 | 0.37 | 0.07 | 0.59 |
| Pair$_4$ | G/14, L14 | **0.86** | 0.61 | **0.24** | 0.94 | 0.07 | 0.02 | 0.00 | 0.06 | 0.82 | **0.64** | **0.23** | **0.92** |

Table 6: Ablation on two-model surrogate sets. Bold numbers are the best in each column; underlined numbers are the second-best.

| Variant | Surrogate Set | GPT-4o | | | | Claude 3.7-extended | | | | Gemini 2.5-Pro | | | |
|---|---|---|---|---|---|---|---|---|---|---|---|---|---|
| | | KMR$_a$ | KMR$_b$ | KMR$_c$ | ASR | KMR$_a$ | KMR$_b$ | KMR$_c$ | ASR | KMR$_a$ | KMR$_b$ | KMR$_c$ | ASR |
| PE$_1$ | B/16, B/32, L32, L16 | 0.87 | 0.65 | 0.26 | **0.99** | 0.54 | 0.32 | 0.07 | 0.68 | 0.80 | 0.57 | 0.16 | 0.90 |
| PE$_2$ | Dino-B, B/32, L32, G/14 | 0.87 | 0.69 | 0.28 | 0.97 | 0.56 | 0.37 | 0.09 | 0.65 | **0.88** | 0.71 | 0.22 | 0.93 |
| PE$_3$ | L16, B/32, L32, G/14 | 0.85 | 0.65 | 0.23 | **0.99** | **0.57** | 0.40 | 0.09 | **0.73** | 0.84 | 0.61 | 0.19 | 0.93 |
| PE$_4$ | B/16, B/32, L32, Dino-B | 0.89 | 0.67 | 0.19 | 0.98 | 0.55 | **0.41** | 0.07 | 0.63 | 0.87 | 0.67 | **0.23** | 0.96 |
| PE$_5$ | B/16, B/32, L32, Dino-S | 0.90 | 0.72 | 0.25 | 0.97 | 0.48 | 0.33 | 0.08 | 0.59 | 0.83 | 0.63 | 0.17 | 0.90 |
| **PE$^+$ (Ours)** | B/16, B/32, L32, G/14 | **0.91** | **0.78** | **0.40** | **0.99** | 0.56 | 0.32 | **0.11** | 0.67 | 0.87 | 0.72 | 0.22 | **0.97** |

Table 7: Ablation on surrogate-set selection. Each row swaps one model in or out of a four-model ensemble. The fully grey PE$^+$ line is our final patch-diverse surrogate set (*CLIP$^\dagger$-G/14, CLIP-B/16, CLIP-B/32, CLIP$^\dagger$-B/32*). Bold numbers denote the best score in each metric column across all variants, underline denote second best with neglectable gap of 0.01

# E    ABLATION STUDY

## E.1    ABLATION STUDY FOR STEP SIZE

This section provides an ablation study for the step size parameter $\alpha$ to view its impact on the performance. Overall, selecting $\alpha \in [0.5, 1.0]$ provides better performance for SSA-CWA, M-Attack. Our M-Attack-V2 prefer stepsize at 1.275, since it adopts ADAM as optimizer.

| Surrogate (paper notation) | Implementation (HuggingFace identifier) |
|---|---|
| CLIP$^\dagger$-B/32 (Ilharco et al., 2021; Schuhmann et al., 2022) | `laion/CLIP-ViT-B-32-laion2B-s34B-b79K` |
| CLIP$^\dagger$-H/14 (Ilharco et al., 2021; Schuhmann et al., 2022) | `laion/CLIP-ViT-H-14-laion2B-s32B-b79K` |
| CLIP-L/14 (Radford et al., 2021) | `openai/clip-vit-large-patch14` |
| CLIP$^\dagger$-B/16 (Ilharco et al., 2021; Schuhmann et al., 2022) | `laion/CLIP-ViT-B-16-laion2B-s34B-b88K` |
| CLIP$^\dagger$-BG/14 (Ilharco et al., 2021; Schuhmann et al., 2022) | `laion/CLIP-ViT-bigG-14-laion2B-39B-b160k` |
| Dino-Small (Oquab et al., 2023) | `facebook/dinov2-small` |
| Dino-Base (Oquab et al., 2023) | `facebook/dinov2-base` |
| Dino-Large (Oquab et al., 2023) | `facebook/dinov2-large` |
| BLIP-2 (2.7 B) (Li et al., 2023) | `Salesforce/blip2-opt-2.7b` |

Table 8: Surrogate models and their corresponding HuggingFace identifier in our main paper.

| $\alpha$ | Method | GPT-4o | | | | Claude 3.7-thinking | | | | Gemini 2.5-Pro | | | |
|---|---|---|---|---|---|---|---|---|---|---|---|---|---|
| | | $KMR_a$ | $KMR_b$ | $KMR_c$ | ASR | $KMR_a$ | $KMR_b$ | $KMR_c$ | ASR | $KMR_a$ | $KMR_b$ | $KMR_c$ | ASR |
| 0.25 | SSA-CWA (Dong et al., 2023a) | 0.08 | 0.08 | 0.04 | 0.10 | 0.06 | 0.03 | 0.00 | 0.03 | 0.06 | 0.03 | 0.00 | 0.01 |
| | M-Attack (Li et al., 2025) | 0.62 | 0.39 | 0.09 | 0.71 | 0.12 | 0.03 | 0.01 | 0.16 | 0.55 | 0.33 | 0.08 | 0.55 |
| | M-Attack-V2 (Ours) | 0.86 | 0.61 | 0.21 | 0.96 | 0.43 | 0.28 | 0.5 | 0.52 | 0.82 | 0.29 | 0.18 | 0.89 |
| 0.50 | SSA-CWA (Dong et al., 2023a) | 0.10 | 0.10 | 0.04 | 0.07 | 0.08 | 0.04 | 0.00 | 0.05 | 0.09 | 0.05 | 0.00 | 0.04 |
| | M-Attack (Li et al., 2025) | 0.73 | 0.48 | 0.17 | 0.77 | 0.20 | 0.13 | 0.06 | 0.22 | 0.79 | 0.53 | 0.10 | 0.80 |
| | M-Attack-V2 (Ours) | 0.87 | 0.64 | 0.23 | 0.96 | 0.58 | 0.34 | 0.13 | 0.67 | 0.83 | 0.59 | 0.17 | 0.94 |
| 1.00 | SSA-CWA (Dong et al., 2023a) | 0.11 | 0.06 | 0.00 | 0.09 | 0.06 | 0.04 | 0.01 | 0.12 | 0.05 | 0.03 | 0.01 | 0.08 |
| | M-Attack (Li et al., 2025) | 0.82 | 0.54 | 0.13 | 0.95 | 0.31 | 0.21 | 0.04 | 0.37 | 0.81 | 0.57 | 0.15 | 0.83 |
| | M-Attack-V2 (Ours) | 0.92 | 0.77 | 0.42 | 0.98 | 0.55 | 0.36 | 0.08 | 0.67 | 0.85 | 0.73 | 0.22 | 0.98 |
| 1.275 | SSA-CWA (Dong et al., 2023a) | 0.09 | 0.09 | 0.04 | 0.03 | 0.06 | 0.03 | 0.00 | 0.03 | 0.05 | 0.02 | 0.00 | 0.02 |
| | M-Attack (Li et al., 2025) | 0.00 | 0.00 | 0.00 | 0.00 | 0.25 | 0.18 | 0.06 | 0.34 | 0.85 | 0.55 | 0.19 | 0.84 |
| | M-Attack-V2 (Ours) | 0.91 | 0.78 | 0.40 | 0.99 | 0.56 | 0.32 | 0.11 | 0.67 | 0.87 | 0.72 | 0.22 | 0.97 |

Table 9: Ablation study on the impact of perturbation budget ($\alpha$).

## E.2 Ablation Study on MCA and ATA Hyperparameters

Fig. 7(left) shows transferability peaks around $K = 10 \sim 20$, beyond which increased stability reduces beneficial noise regularization. Fig. 7(right) demonstrates larger $\lambda$ boosts diversity by aligning semantics closer to auxiliary data but risks impairing semantic accuracy (as measured by KMR). Fig. 8(a,b) indicates minor impacts from $P$ and momentum coefficient $\beta$; setting $P = 2$ optimizes performance and efficiency, and the default $\beta = 0.9$ consistently yields robust results.

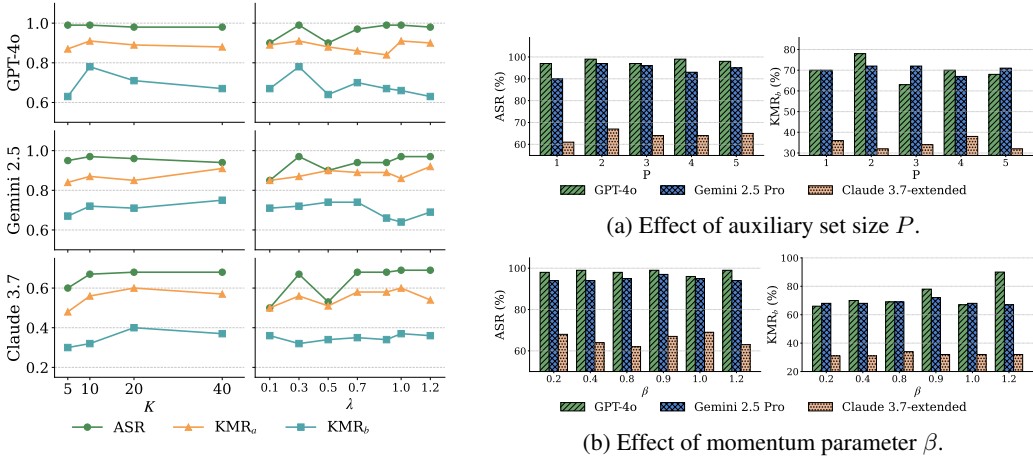

(a) Effect of auxiliary set size $P$.

(b) Effect of momentum parameter $\beta$.

Figure 7: ASR and $KMR_a$/$KMR_b$ vs. different $K$ and $\lambda$.

Figure 8: Ablation study on auxiliary set size $P$ and momentum parameter $\beta$.

## F Additional Results

### F.1 Additional Results on 1K image

We compare M-Attack and M-Attack-V2 on 1K images for better statistical stability. We changed the threshold into multiple values since no additional keywords were added for the 900 images, thus replacing the KMR with ASR with thresholds at different matching levels. Our M-Attack-V2 achieves consistently better results compared to M-Attack, showing superiority of our proposed strategy.

| threshold | GPT-4o | | Gemini-2.5-Pro | | Claude-3.7-extended | |
|---|---|---|---|---|---|---|
| | M-Attack | M-Attack-V2 | M-Attack | M-Attack-V2 | M-Attack | M-Attack-V2 |
| 0.3 | 0.868 | 0.983 | 0.714 | 0.915 | 0.289 | 0.632 |
| 0.4 | 0.614 | 0.965 | 0.621 | 0.870 | 0.250 | 0.437 |
| 0.5 | 0.614 | 0.871 | 0.539 | 0.673 | 0.057 | 0.127 |
| 0.6 | 0.399 | 0.423 | 0.310 | 0.556 | 0.015 | 0.127 |
| 0.7 | 0.399 | 0.412 | 0.245 | 0.342 | 0.013 | 0.089 |
| 0.8 | 0.234 | 0.328 | 0.230 | 0.289 | 0.008 | 0.009 |
| 0.9 | 0.056 | 0.150 | 0.049 | 0.087 | 0.001 | 0.005 |

Table 10: Comparison of results on 1K images. We provide ASR based on different thresholds as a surrogate for KMR following `M-Attack` (Li et al., 2025).

## F.2 ADDITIONAL RESULTS ON FGSM FRAMEWORK

We provide the results of the I-FGSM (Kurakin et al., 2018) and MI-FGSM (Dong et al., 2018) under our `M-Attack` framework as complementary, presented in Tab. 15. Results show that even under the FGSM framework, where the patchy gradient matter is smoothed by assigning $\text{sign}(\nabla \mathcal{L})$, `M-Attack-V2` still benefit from momentum. Moreover, MI-FGSM still provides results comparable to those of the ADAM version. However, using PGD framework with ADAM optimizer is generally the better choice to unleash the potential of black-box attack fully since it can better explore in the space while also reducing scale issue with second-order momentum.

## F.3 ADDITIONAL RESULTS ON OPEN-SOURCE MLLMS

We extend the evaluation from black-box commercial MLLMs to two open-source MLLMs, Qwen-2.5-VL (Bai et al., 2025) and LLaVa-1.5 (Liu et al., 2024). The setting follows exactly the same as in the main paper. Results in Tab. 11 shows that our method consistently achieves the best result compared to other method on both commercial black-box models and open-source white-box models.

| Method | Qwen-2.5-VL | | | | LLaVA-1.5 | | | |
|---|---|---|---|---|---|---|---|---|
| | $KMR_a$ | $KMR_b$ | $KMR_c$ | ASR | $KMR_a$ | $KMR_b$ | $KMR_c$ | ASR |
| AttackVLM | 0.12 | 0.04 | 0.00 | 0.01 | 0.11 | 0.03 | 0.00 | 0.07 |
| SSA-CWA | 0.36 | 0.25 | 0.04 | 0.38 | 0.29 | 0.17 | 0.04 | 0.34 |
| AnyAttack | 0.53 | 0.28 | 0.09 | 0.53 | 0.60 | 0.32 | 0.07 | 0.58 |
| FOA-Attack | 0.83 | 0.61 | 0.20 | 0.91 | 0.94 | 0.75 | **0.29** | 0.95 |
| M-Attack | 0.80 | 0.65 | 0.17 | 0.90 | 0.85 | 0.59 | 0.20 | 0.95 |
| M-Attack-V2 | **0.87** | **0.67** | **0.27** | **0.95** | **0.96** | **0.83** | **0.29** | **0.96** |

Table 11: Comparison on Qwen-2.5-VL and LLaVA-1.5. Higher $KMR_{a/b/c}$ and ASR are better. Best results are bold.

## F.4 ADDITIONAL RESULTS ON OTHER GPT-5 REASONING MODES

GPT-5 provides four reasoning modes: *minimum*, *low*, *medium*, and *high*. While the main paper presents results using GPT-5-thinking-*low*, additional experiments on other reasoning modes are summarized in Tab. 12. Our proposed `M-Attack-V2` consistently achieves superior performance across all modes. Interestingly, providing additional thinking budget generally enhances model robustness, evidenced by a reduction in ASR and KMR. However, this improvement is not strictly monotonic: ASR first decreases from 100% (*low*) to 96% (*medium*) before slightly rebounding to 99% (*high*). A similar non-monotonic trend can also be observed elsewhere in the table.

## F.5 CROSS-DOMAIN EVALUATION ON MEDICAL AND OVERHEAD IMAGERY

Beyond the general-domain datasets, we further probe transferability to domains that are notoriously challenging for closed-source VLMs: chest X-rays and overhead remote sensing. Concretely, we augment the *NIPS 2017 adversarial competition* evaluation with images from *ChestMNIST*, from *MedMNIST* (Yang et al., 2021) and *PatternNet* (Li et al., 2018). We keep the target set unchanged and

| Method | Model | GPT-5 (low) | | | | GPT-5 (medium) | | | | GPT-5 (high) | | | |
|---|---|---|---|---|---|---|---|---|---|---|---|---|---|
| | | $KMR_a$ | $KMR_b$ | $KMR_c$ | ASR | $KMR_a$ | $KMR_b$ | $KMR_c$ | ASR | $KMR_a$ | $KMR_b$ | $KMR_c$ | ASR |
| SSA-CWA (Dong et al., 2023a) | Ensemble | 0.08 | 0.04 | 0.00 | 0.08 | 0.09 | 0.05 | 0.01 | 0.06 | 0.10 | 0.05 | 0.01 | 0.07 |
| FOA-Attack (Jia et al., 2025) | Ensemble | 0.90 | 0.67 | 0.23 | 0.94 | 0.90 | 0.69 | 0.21 | **0.96** | 0.87 | 0.68 | 0.24 | 0.96 |
| M-Attack (Li et al., 2025) | Ensemble | 0.89 | 0.65 | 0.25 | 0.98 | 0.85 | 0.61 | 0.16 | **0.96** | 0.80 | 0.60 | 0.20 | 0.93 |
| M-Attack-V2 (Ours) | Ensemble | **0.92** | **0.79** | **0.30** | **1.00** | **0.90** | **0.73** | **0.25** | **0.96** | **0.88** | **0.71** | **0.27** | **0.99** |

Table 12: Comparison on GPT-5 under three budget settings (low/medium/high).

reuse the same attack budget and optimization hyper-parameters as in the main experiments. These domains are non-photographic and typically elicit generic captions from off-the-shelf VLMs, making them a stringent test of cross-domain transfer.

We report $KMR_a/KMR_b/KMR_c$ and ASR (higher is better) on GPT-4o, Claude 3.7, and Gemini 2.5 in Tables 13 and 14. Across both datasets, M-Attack-V2 consistently surpasses M-Attack and prior baselines. On *PatternNet*, M-Attack-V2 improves Claude 3.7 ASR from 0.48 to 0.73 (+0.25) and raises GPT-4o $KMR_{a/b/c}$ to 0.83/0.71/0.24. On *ChestMNIST*, the gains are even larger on Claude 3.7 (ASR $0.31 \rightarrow 0.83$, +0.52), while M-Attack-V2 also achieves the highest $KMR_{a/b/c}$ on Gemini 2.5 (0.89/0.76/0.33). The only exception is ChestMNIST ASR on Gemini 2.5, where M-Attack is marginally higher (0.96 vs. 0.95), despite M-Attack-V2 yielding stronger keyword-match rates.

| Method | GPT-4o | | | | Claude 3.7 | | | | Gemini 2.5 | | | |
|---|---|---|---|---|---|---|---|---|---|---|---|---|
| | $KMR_a$ | $KMR_b$ | $KMR_c$ | ASR | $KMR_a$ | $KMR_b$ | $KMR_c$ | ASR | $KMR_a$ | $KMR_b$ | $KMR_c$ | ASR |
| AttackVLM | 0.06 | 0.01 | 0.00 | 0.02 | 0.06 | 0.02 | 0.00 | 0.00 | 0.09 | 0.04 | 0.00 | 0.03 |
| SSA-CWA | 0.05 | 0.02 | 0.00 | 0.13 | 0.04 | 0.03 | 0.00 | 0.07 | 0.08 | 0.02 | 0.01 | 0.15 |
| AnyAttack | 0.06 | 0.03 | 0.00 | 0.05 | 0.03 | 0.01 | 0.00 | 0.05 | 0.06 | 0.02 | 0.00 | 0.05 |
| M-Attack | 0.79 | 0.66 | 0.21 | **0.93** | 0.33 | 0.17 | 0.04 | 0.48 | 0.86 | **0.71** | **0.23** | 0.91 |
| M-Attack-V2 | **0.83** | **0.71** | **0.24** | **0.93** | **0.58** | **0.40** | **0.09** | **0.73** | **0.88** | 0.68 | 0.22 | **0.97** |

Table 13: Cross-domain results on *PatternNet* (Li et al., 2018). We report $KMR_a/KMR_b/KMR_c$ and ASR (higher is better). **Bold** = best in column; underline = second best. The shaded row is our method.

| Method | GPT-4o | | | | Claude 3.7 | | | | Gemini 2.5 | | | |
|---|---|---|---|---|---|---|---|---|---|---|---|---|
| | $KMR_a$ | $KMR_b$ | $KMR_c$ | ASR | $KMR_a$ | $KMR_b$ | $KMR_c$ | ASR | $KMR_a$ | $KMR_b$ | $KMR_c$ | ASR |
| AttackVLM | 0.06 | 0.01 | 0.00 | 0.03 | 0.05 | 0.02 | 0.00 | 0.02 | 0.08 | 0.03 | 0.00 | 0.02 |
| SSA-CWA | 0.06 | 0.03 | 0.00 | 0.15 | 0.04 | 0.03 | 0.00 | 0.07 | 0.08 | 0.02 | 0.01 | 0.14 |
| AnyAttack | 0.06 | 0.02 | 0.00 | 0.05 | 0.03 | 0.01 | 0.00 | 0.04 | 0.07 | 0.02 | 0.00 | 0.05 |
| M-Attack-V1 | 0.89 | 0.70 | 0.22 | 0.92 | 0.31 | 0.18 | 0.07 | 0.31 | 0.85 | 0.67 | 0.23 | **0.96** |
| M-Attack-V2 | **0.90** | **0.74** | **0.27** | **0.97** | **0.70** | **0.51** | **0.21** | **0.83** | **0.89** | **0.76** | **0.33** | 0.95 |

Table 14: Cross-domain results on *ChestMNIST*, from MedMNIST (Yang et al., 2021). We report $KMR_a/KMR_b/KMR_c$ and ASR (higher is better). Bold = best in column; underline = second best. The shaded row is our method.

## F.6 ROBUSTNESS TO INPUT–PREPROCESSING DEFENSES

We evaluate two input–preprocessing defenses—JPEG recompression (quality $Q=75$) and diffusion-based purification (DiffPure) with denoising budgets $t=25$ and $t=75$. As summarized in Table 16, the JPEG results show that M-Attack-V2 remains strong while prior attacks substantially degrade, suggesting resilience to quantization and mild photometric shifts. DiffPure reduces success rates for all methods; however, M-Attack-V2 preserves a clear margin at $t=25$ and remains the most effective even under the aggressive $t=75$ setting, where purification approaches image regeneration.

| Method | Model | GPT-4o | | | | Claude 3.7-extended | | | | Gemini 2.5-Pro | | | |
|---|---|---|---|---|---|---|---|---|---|---|---|---|---|
| | | KMR$_a$ | KMR$_b$ | KMR$_c$ | ASR | KMR$_a$ | KMR$_b$ | KMR$_c$ | ASR | KMR$_a$ | KMR$_b$ | KMR$_c$ | ASR |
| M-Attack-V2-ADAM (Ours) | Ensemble | 0.91 | 0.78 | 0.40 | 0.99 | 0.56 | 0.32 | 0.11 | 0.67 | 0.87 | 0.72 | 0.22 | 0.97 |
| M-Attack-V2-FGSM | Ensemble | 0.85 | 0.64 | 0.19 | 0.98 | 0.40 | 0.26 | 0.08 | 0.46 | 0.83 | 0.65 | 0.17 | 0.90 |
| M-Attack-V2-MIFGSM | Ensemble | 0.90 | 0.66 | 0.23 | 0.96 | 0.45 | 0.30 | 0.07 | 0.57 | 0.84 | 0.64 | 0.15 | 0.87 |

Table 15: Ablation study of M-Attack-V2 under different optimizer/attack variants.

| Setting | Method | GPT-4o | | | | Claude 3.7 | | | | Gemini 2.5 | | | |
|---|---|---|---|---|---|---|---|---|---|---|---|---|---|
| | | KMR$_a$ | KMR$_b$ | KMR$_c$ | ASR | KMR$_a$ | KMR$_b$ | KMR$_c$ | ASR | KMR$_a$ | KMR$_b$ | KMR$_c$ | ASR |
| **JPEG** ($Q$=75) | AttackVLM | 0.06 | 0.02 | 0.00 | 0.03 | 0.07 | 0.02 | 0.00 | 0.02 | 0.08 | 0.04 | 0.00 | 0.04 |
| | SSA-CWA | 0.08 | 0.04 | 0.01 | 0.10 | 0.07 | 0.02 | 0.00 | 0.05 | 0.09 | 0.06 | 0.01 | 0.09 |
| | AnyAttack | 0.06 | 0.03 | 0.00 | 0.05 | 0.04 | 0.01 | 0.00 | 0.03 | 0.08 | 0.03 | 0.00 | 0.05 |
| | M-Attack | 0.76 | 0.54 | 0.16 | 0.91 | 0.28 | 0.17 | 0.03 | 0.34 | **0.75** | 0.51 | 0.11 | 0.76 |
| | M-Attack-V2 | **0.89** | **0.69** | **0.20** | **0.97** | **0.55** | **0.36** | **0.09** | **0.68** | **0.75** | **0.56** | **0.18** | **0.82** |
| **DiffPure** ($t$=25) | AttackVLM | 0.05 | 0.02 | 0.00 | 0.01 | 0.05 | 0.02 | 0.00 | 0.01 | 0.08 | 0.03 | 0.00 | 0.01 |
| | SSA-CWA | 0.07 | 0.03 | 0.00 | 0.02 | 0.04 | 0.02 | 0.00 | 0.03 | 0.07 | 0.01 | 0.00 | 0.05 |
| | AnyAttack | 0.07 | 0.03 | 0.00 | 0.04 | 0.02 | 0.02 | 0.00 | 0.04 | 0.09 | 0.04 | 0.00 | 0.07 |
| | M-Attack | 0.42 | 0.20 | 0.03 | 0.43 | 0.10 | 0.05 | 0.01 | 0.10 | 0.39 | 0.22 | 0.01 | 0.32 |
| | M-Attack-V2 | **0.73** | **0.47** | **0.15** | **0.72** | **0.19** | **0.11** | **0.04** | **0.20** | **0.61** | **0.42** | **0.06** | **0.56** |
| **DiffPure** ($t$=75) | AttackVLM | 0.08 | 0.05 | 0.00 | 0.02 | 0.04 | 0.02 | 0.00 | 0.00 | 0.04 | 0.01 | 0.00 | 0.01 |
| | SSA-CWA | 0.05 | 0.03 | **0.01** | 0.06 | 0.05 | **0.03** | 0.00 | 0.03 | 0.07 | 0.02 | 0.00 | 0.05 |
| | AnyAttack | 0.05 | 0.00 | 0.00 | 0.06 | 0.04 | 0.02 | 0.00 | 0.03 | 0.04 | 0.02 | 0.00 | 0.07 |
| | M-Attack | 0.10 | 0.02 | 0.00 | 0.04 | 0.03 | 0.02 | 0.00 | 0.02 | 0.05 | 0.05 | 0.00 | 0.05 |
| | M-Attack-V2 | **0.13** | **0.06** | **0.01** | **0.07** | **0.07** | 0.02 | 0.00 | **0.06** | **0.12** | **0.06** | **0.01** | **0.08** |

Table 16: Unified robustness under input–preprocessing defenses. We report KMR$_a$, KMR$_b$, KMR$_c$, and ASR (↑) for GPT-4o, Claude-3.7, and Gemini-2.5. Bold indicates the best value within each metric column for the given defense block; shaded cells highlight M-Attack-V2 (numeric cells only).

# G VISUALIZATION

## G.1 VISUALIZATION OF ADVERSARIAL SAMPLES

Fig. 9 and Fig. 10 visualize adversarial samples of different black-box attack algorithms under different perturbation constraints. Under $\epsilon = 8$, no significant difference exists between M-Attack and M-Attack-V2. On the $\epsilon = 16$ setting, the visual effect is still very close between M-Attack and M-Attack-V2. Since our M-Attack-V2 also greatly improves the results under $\epsilon = 8$, future directions might be improving the imperceptibility by adding constraints besides the $\ell_\infty$. We also provide all 100 images in the supplementary martial for further reference.

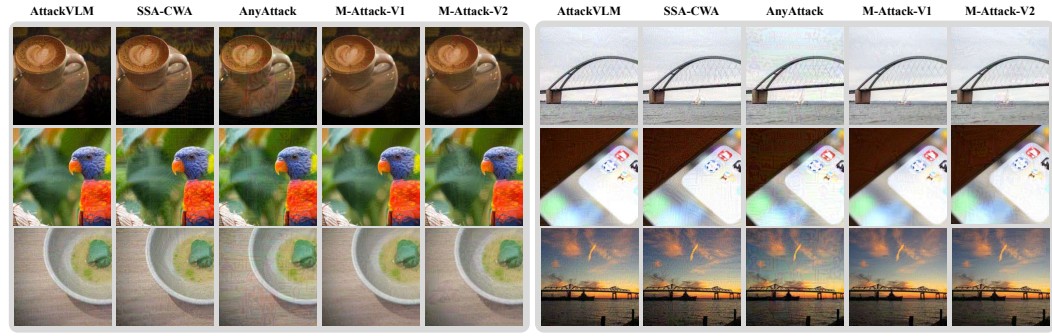

Figure 9: Visualization of adversarial samples under $\epsilon = 8$.

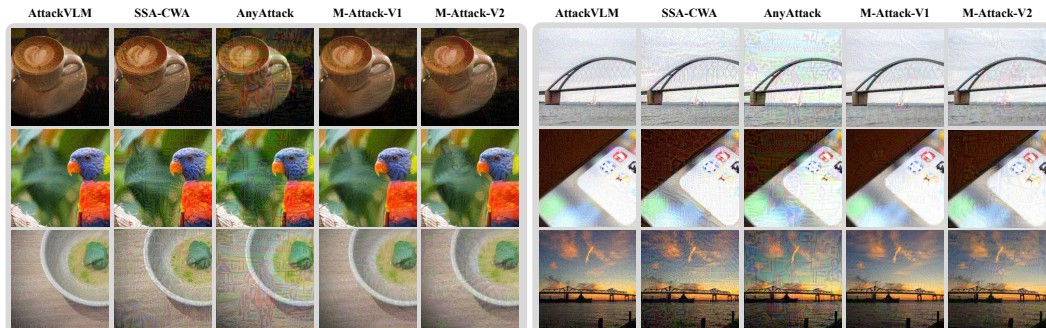

Figure 10: Visualization of adversarial samples under $\epsilon = 16$.

### G.2 VISUALIZATION OF REASONING MODELS

Fig. 11 illustrates how GPT-o3 (OpenAI, 2025) responds to our adversarial samples. The model's visual reasoning behaviors can be broadly categorized into three types: *no reasoning* (response (d)), *simple reasoning* (responses (b) and (c)), and *zoom-in reasoning* (response (a)). Notably, in response (a), GPT-o3 already identifies the central area as uncertain and zooms in on it. However, its reasoning mechanism is not well-equipped to handle adversarial perturbations, resulting in a response that remains semantically close to the target image despite the perturbation. This observation suggests that vision reasoning offers a degree of robustness by detecting uncertainty and taking subsequent actions. During training, incorporating explicit behaviors, such as refusing to answer or flagging potential adversarial inputs, could further enhance the utility of vision-based inference under adversarial conditions.

## H DISCUSSION

### H.1 LIMITATION

Despite the strong and state-of-the-art attacking performance on various closed-source MLLMs, the proposed M-Attack-V2 still relies on surrogate model ensembles and fine-grained visual alignment strategies, which may limit its applicability in extreme cases and domains where high-fidelity surrogate models or visual data are unavailable. The method also assumes some degree of consistency and diversity among surrogate model representations, which might not hold across all different architectures or domain-shifted datasets. Moreover, while the attack improves transferability, it may require slightly extra computational resources for more ensembles during optimization. Future work will explore efficiency-aware variants and more generalizable attack strategies beyond current assumptions of semantic alignment.

### H.2 BORDER IMPACT

The development of M-Attack-V2 advances our understanding of the vulnerabilities in LVLMs under black-box settings, particularly in real-world, security-critical applications. By enabling fine-grained detail targeting and significantly improving attack success rates without access to model internals, this work highlights the risks posed by adversarial manipulation to commercial systems used in autonomous driving, content generation, medical imaging, etc. These insights can guide the design of more robust LVLMs and encourage the community to adopt stronger evaluation protocols and defense mechanisms. Additionally, M-Attack-V2 serves as a valuable benchmark for future research on secure multimodal AI, encouraging the development of resilient architectures that are better aligned with societal safety and reliability standards.

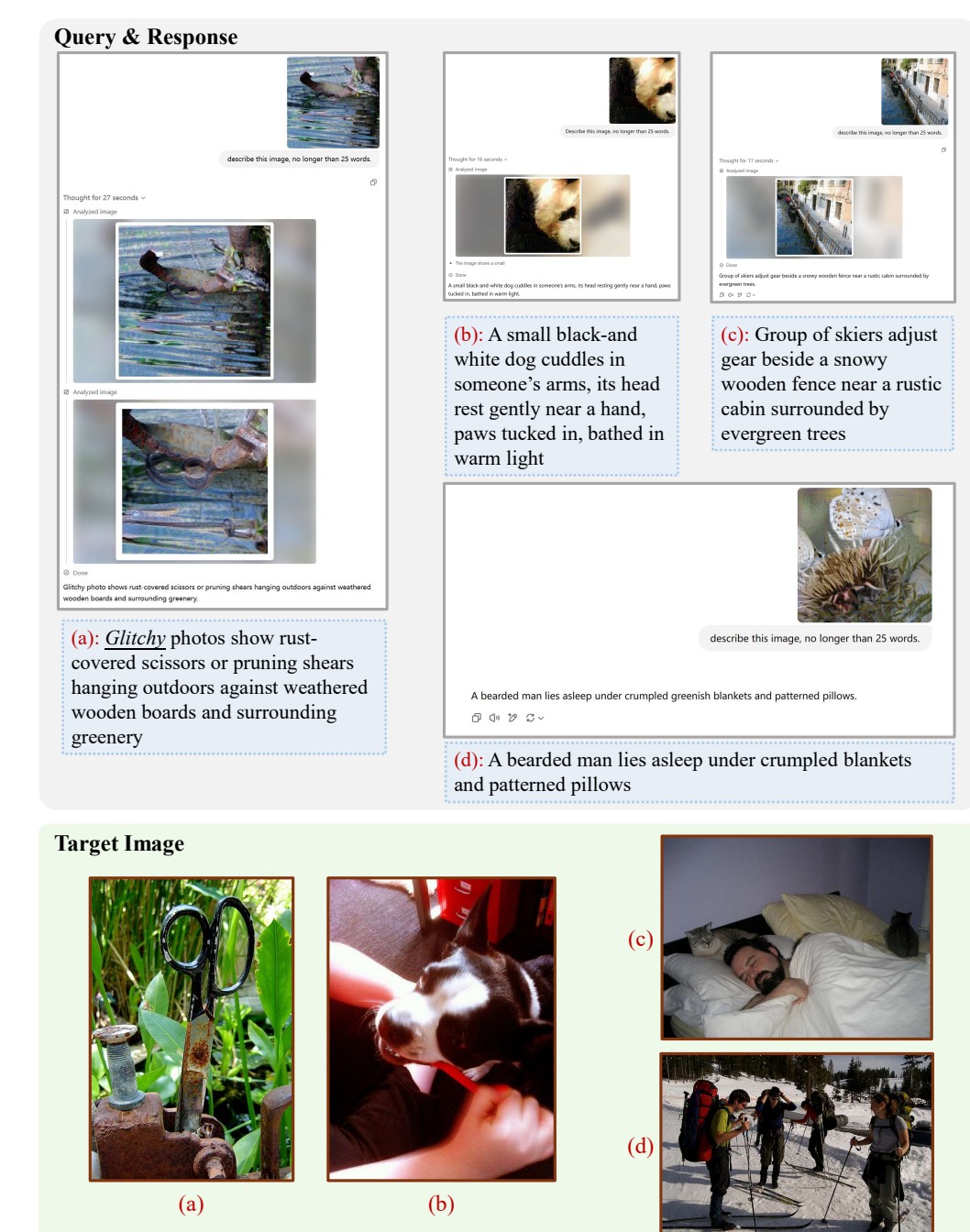

Figure 11: Visualization of GPT-o3's response towards `M-Attack-V2` adversarial samples. The underlined 'glitchy' denotes that O3 notices something unusual.

# I  USE OF LARGE LANGUAGE MODELS

We utilize Large Language Models (LLMs) to refine portions of writing for our manuscript, but not to generate research ideas. Additionally, following the *LLM as Judge* evaluation paradigm (Zheng et al., 2023) and the exact setup described in `M-Attack` (Li et al., 2025), we utilize GPT-4o from the OpenAI API for our standard evaluation of KMR and ASR metrics. The prompts and parameters used are identical to those specified in `M-Attack`, thereby ensuring complete reproducibility.

