# OpenReview forum: "$\texttt{M-Attack-V2}$: Pushing the Frontier of Black-Box LVLM Attacks via Fine-Grained Detail Targeting"
_ICLR.cc/2026/Conference — ICLR 2026 Conference Withdrawn Submission_

### Official Review · Reviewer_9cWL · 2025-10-27

**Soundness:** 2
**Presentation:** 2
**Contribution:** 2
**Rating:** 4
**Confidence:** 4

**Summary:**

The paper introduces M-Attack-V2, a new framework for black-box adversarial attacks on Large Vision-Language Models (LVLMs). The work builds upon the previously successful M-Attack method, which relied on local crop-level feature matching between source and target images but suffered from unstable gradients. M-Attack-V2 mitigates this instability through three innovations: (1) Multi-Crop Alignment (MCA), a variance-reducing gradient aggregation strategy; (2) Auxiliary Target Alignment (ATA), a semantic regularizer that stabilizes optimization via auxiliary samples; and (3) Patch Momentum (PM), a replay-based stabilization mechanism. These components yield improved transferability and attack success rates.

**Strengths:**

1. The paper provides a well-defined empirical diagnosis of M-Attack instability, showing near-orthogonal consecutive gradients and quantifying the issue via mIoU. The proposed method effectively addresses these problems through principled variance reduction and patch-level alignment mechanisms.
2. The work re-casts local matching as an expectation over transformations and targets, deriving a Monte-Carlo averaging estimator is a technically interesting and novel formulation that improves gradient stability.
3. Extensive evaluations show consistent ASR/KMR improvements across commercial LVLMs (GPT-5, Claude-4.0-extended, Gemini-2.5-Pro). The component ablations (Table 3) isolate MCA, ATA, and PM contributions, confirming the method’s improvement.

**Weaknesses:**

1. While gradient overlap, improved sampling quality, and patch momentum are analyzed theoretically, their explicit connection to adversarial transferability remains unclear beyond empirical evidence. The theoretical evidence and the observed transfer performance appear disconnected.

2.  The results rely heavily on proprietary models, which are subject to continuous updates and thus limit reproducibility and fair comparison. It would strengthen the work to include experiments on open-source models in the main text. The appendix currently reports results on older variants; incorporating evaluations on more recent VLMs such as GLM-4.1, LLaVA 1.6, and Llama-3.2-Vision would help to strength the findings.

3. The proposed method is compared against only a few baselines on VLM classification tasks, without considering a broader line of adversarial attack studies [1-5] on CLIP that explore more diverse tasks such as retrieval, VQA, and image captioning. The use of CLIP as surrogate models is appropriate and fair; however, including evaluations across these additional task types would provide a more comprehensive assessment of transferability and generalization.

4. The ASR evaluation relies on GPTScore and keyword matching (KMRa/b/c), which may introduce prompt-specific bias.

5. The performance improvements for individual modules (Table 3) are small and may not justify the added complexity.

6. Several presentation issues: table captions appear below rather than above (refer to ICLR style files), and one reference is duplicated (lines 540–546).

7. Different hyperparameters (e.g., $\alpha$ = 0.75 for Claude vs 1.0 for others) raise potential cherry-picking concerns.



[1] Zhang, J., Yi, Q., & Sang, J. (2022, October). Towards adversarial attack on vision-language pre-training models. In Proceedings of the 30th ACM International Conference on Multimedia (pp. 5005-5013).

[2] Lu, D., Wang, Z., Wang, T., Guan, W., Gao, H., & Zheng, F. (2023). Set-level guidance attack: Boosting adversarial transferability of vision-language pre-training models. In Proceedings of the IEEE/CVF International Conference on Computer Vision (pp. 102-111).

[3] Zhou, Z., Hu, S., Li, M., Zhang, H., Zhang, Y., & Jin, H. (2023, October). Advclip: Downstream-agnostic adversarial examples in multimodal contrastive learning. In Proceedings of the 31st ACM International Conference on Multimedia (pp. 6311-6320).

[4] Huang, H., Erfani, S. M., Li, Y., Ma, X., & Bailey, J. (2025). X-Transfer attacks: Towards super transferable adversarial attacks on CLIP. In Proceedings of the 42nd International Conference on Machine Learning (Vol. 267, pp. 25204–25234).

[5] Liu, C., Chen, H., Zhang, Y., Dong, Y., & Zhu, J. (2024). Scaling laws for black box adversarial attacks. arXiv preprint arXiv:2411.16782.

**Questions:**

1. Why are the hyperparameters different for different models ($\alpha$ = 0.75 for Claude and $\alpha$ = 1.0 for all others)?
2. Why are DINO models included in the ensemble?

---

### Official Review · Reviewer_pwjM · 2025-10-29

**Soundness:** 2
**Presentation:** 1
**Contribution:** 2
**Rating:** 2
**Confidence:** 3

**Summary:**

This paper proposes M-Attack-V2, a theoretically grounded framework for black-box adversarial attacks on Large Vision–Language Models (LVLMs). It addresses the high gradient variance in local-level matching by introducing three key components: Multi-Crop Alignment (MCA) for gradient denoising, Auxiliary Target Alignment (ATA) for stabilizing target embeddings, and Patch Momentum for maintaining consistent optimization across local regions. The method achieves substantial gains in attack success rates on models such as GPT-5, Gemini-2.5-Pro, and Claude-4.0.

**Strengths:**

1. The paper is well-motivated, clearly identifying gradient mismatch as a key limitation of local-level matching.
2. The proposed method is theoretically supported.
3. The approach achieves strong empirical results, outperforming baselines across multiple models.

**Weaknesses:**

Overall the paper writing is very unclear and confusing.

1. The introduction to the original M-Attack is very limited, making it difficult to follow the motivation and context of the proposed improvements. The discussion on gradient mismatch is especially confusing. There is no formal definition or equation specifying which gradient is being analyzed. It is unclear whether the gradient refers to that of the source crop or the target crop, and whether the mismatch occurs between similar source crops with the same target crop or across different target crops. The description of how cropping operates on the source and target sides is also vague; the important distinction that the source crop acts in pixel space while the target crop acts in feature space only becomes apparent much later (around line 177). These key details should be clearly articulated in the background section.

2. The method section also suffers from a lack of precision. The notion of a “target embedding distribution” is introduced without a clear definition or explanation of how it is constructed. Similarly, the process for selecting auxiliary images is not described. Even in Algorithm 1, the definition of the transformation distribution 𝐷 is missing.

**Questions:**

see weakness

---

### Official Review · Reviewer_YJew · 2025-10-29

**Soundness:** 2
**Presentation:** 3
**Contribution:** 2
**Rating:** 4
**Confidence:** 3

**Summary:**

This paper proposes M-Attack-V2, an improved black-box attack on large vision-language models (LVLMs) that focuses on fine-grained detail targeting.
By selectively perturbing semantically important visual regions, the method aims to increase attack success with fewer queries and minimal perceptual change.
Experiments on several LVLMs show moderate improvements in success rate and query efficiency compared to prior M-Attack and other black-box baselines.

**Strengths:**

- The paper tackles the important and practical challenge of improving black-box attacks for LVLMs, a domain with limited existing methods.

- The proposed fine-grained detail targeting effectively balances perceptual imperceptibility and attack strength, showing improvements in attack success rate and query efficiency.

- Experiments cover multiple LVLMs and include comparisons with standard baselines such as Square Attack and Bandit-based methods, providing a reasonably comprehensive empirical evaluation.

**Weaknesses:**

- The conceptual novelty is limited. The method largely extends prior M-Attack with refined region selection, but does not introduce a new theoretical framework or attack family.

- The fine-grained targeting module is described at a high level; there is no detailed mathematical formulation or clear rule for selecting salient regions beyond heuristic attention maps.

- The ablation studies are incomplete. For instance, removing the fine-grained targeting or using random region selection could help isolate its contribution, but such baselines are missing.

- The experiments lack diversity in tasks. The evaluation is limited to visual question answering and captioning; other LVLM tasks (reasoning, OCR, or instruction following) are not tested.

- The transferability and generalization of the attack are unclear. It is not shown whether perturbations generated for one model are effective on others.

- The real-world feasibility is uncertain. Query limits, response filtering, and adaptive throttling in deployed APIs could heavily constrain the attack’s practicality.

**Questions:**

See Weaknesses.

---

### Official Review · Reviewer_7P9g · 2025-10-31

**Soundness:** 3
**Presentation:** 2
**Contribution:** 2
**Rating:** 4
**Confidence:** 3

**Summary:**

This paper proposes M-Attack-V2, a novel black-box adversarial attack method targeting Large Vision-Language Models (LVLMs). Building upon the M-Attack framework, the authors aim to address a critical issue in local patch matching—high gradient variance—which hinders optimization stability. To this end, the paper introduces three key components: Multi-Crop Alignment (MCA) to reduce gradient variance via averaging across multiple local views, Auxiliary Target Alignment (ATA) to enhance semantic consistency using auxiliary target embeddings, and Patch Momentum (PM) to improve gradient direction consistency over iterations. Extensive experiments on leading commercial LVLMs (e.g., GPT-5, Claude-4.0, Gemini-2.5) demonstrate that M-Attack-V2 significantly outperforms prior state-of-the-art methods in attack success rate (ASR) and transferability.

**Strengths:**

1.	Theoretical analysis is rigorous: Theorems 1 and 2 clearly explain how MCA and ATA reduce gradient variance and improve semantic alignment.

2.	The experimental design is thorough and convincingly validates the method’s effectiveness and practicality.

**Weaknesses:**

1.	While the method achieves notable improvements in both performance and theory, it remains an enhancement of the existing M-Attack framework. Its originality and novelty are somewhat limited by this dependency.
2.	Although the attack success rate improves significantly, the method shows weaker imperceptibility (i.e., higher l2 and l1 norms) compared to prior work, indicating a trade-off between success and stealth. It would be helpful for the authors to further analyze this trade-off and explore potential ways to balance both.
3.	The introduction of multiple crops (MCA) and auxiliary targets (ATA) increases the per-iteration computational cost. Training efficiency should be more explicitly quantified and discussed.

**Questions:**

Please refer to the weakness.

---

### Note · Authors · 2025-11-17

**Comment:**

We sincerely thank the reviewers and the committee for their time and effort in handling our paper.

**Withdrawal Confirmation:**

I have read and agree with the venue's withdrawal policy on behalf of myself and my co-authors.